# Methane stimulates massive nitrogen loss from freshwater reservoirs in India

S. Wajih A. Naqvi[1,2,4], Phyllis Lam[2,3], Gayatree Narvenkar[1], Amit Sarkar[1], Hema Naik[1], Anil Pratihary[1,2], Damodar M. Shenoy[1], Mangesh Gauns[1], Siby Kurian[1], Samir Damare[1], Manon Duret[3], Gaute Lavik[2] & Marcel M.M. Kuypers[2]

The fate of the enormous amount of reactive nitrogen released to the environment by human activities in India is unknown. Here we show occurrence of seasonal stratification and generally low concentrations of dissolved inorganic combined nitrogen, and high molecular nitrogen ($N_2$) to argon ratio, thus suggesting seasonal loss to $N_2$ in anoxic hypolimnia of several dam-reservoirs. However, $^{15}N$-experiments yielded low rates of denitrification, anaerobic ammonium oxidation and dissimilatory nitrate reduction to ammonium—except in the presence of methane ($CH_4$) that caused ~12-fold increase in denitrification. While nitrite-dependent anaerobic methanotrophs belonging to the NC10 phylum were present, previously considered aerobic methanotrophs were far more abundant (up to 13.9%) in anoxic hypolimnion. Methane accumulation in anoxic freshwater systems seems to facilitate rapid loss of reactive nitrogen, with generally low production of nitrous oxide ($N_2O$), through widespread coupling between methanotrophy and denitrification, potentially mitigating eutrophication and emissions of $CH_4$ and $N_2O$ to the atmosphere.

[1] CSIR-National Institute of Oceanography, Dona Paula, Goa 403 004, India. [2] Max-Planck Institute for Marine Microbiology, Celsiusstrasse 1, D-28359 Bremen, Germany. [3] Ocean and Earth Science, National Oceanography Centre Southampton, University of Southampton, European Way, Southampton SO14 3ZH, UK. [4]Present address: Council of Scientific & Industrial Research, Rafi Marg, New Delhi 110 001, India. Correspondence and requests for materials should be addressed to S.W.A.N. (email: wajih_naqvi@yahoo.com)

There is a dearth of information on the biogeochemistry of lakes and reservoirs and how they are impacted by human activities in South Asia, a region that accounts for about a quarter of the world's human population. India is one of the largest consumers of synthetic nitrogen fertilisers in the world (~17 million tons N per year)[1]. Altogether with reactive nitrogen release from other sources (e.g., fossil fuel combustion and waste disposal), anthropogenic nitrogen is predicted to result in the eutrophication of aquatic bodies including the coastal ocean[2] and thus intensification of seasonal hypoxia to anoxia. On the one hand, intensified anoxia facilitates greater loss of reactive nitrogen through microbial reduction of nitrate ($NO_3^-$) to molecular nitrogen ($N_2$). On the other hand, it promotes the production of two potent greenhouse gases–methane ($CH_4$) and nitrous oxide ($N_2O$). Nitrogen loss from anaerobic aquatic environments had long been believed to occur through heterotrophic denitrification ($NO_3^- \rightarrow NO_2^- \rightarrow NO \rightarrow N_2O \rightarrow N_2$), until anaerobic ammonium oxidation (anammox) ($NO_2^- + NH_4^+ \rightarrow N_2 + 2H_2O$) was discovered to be an important component of the nitrogen cycle[3,4]. More recently, nitrite-dependent anaerobic methane oxidation (N-DAMO) ($3CH_4 + 8NO_2^- + 8H^+ \rightarrow 3CO_2 + 4N_2 + 10H_2O$) has been found as yet another $N_2$ production pathway[5]. This is potentially an environmentally significant process not only because of its role in nitrogen loss, but also due to its consumption of $CH_4$ and constraints on the formation of $N_2O$, which would otherwise be produced through canonical denitrification[6]. However, potential of N-DAMO has so far largely been inferred from enrichment and molecular detection in environmental samples of the bacteria known to mediate this process – 'Candidatus Methylomirabilis oxyfera'[5] and its relatives in the phylum NC10[7–16]. All field studies conducted so far have focussed on soils and sediments with just a few exceptions. In one case, NC10 bacteria have been reported from the water column of a dam-reservoir[11]; while in two other studies they were found within the oceanic oxygen minimum zone (OMZ) of the eastern tropical North Pacific[17,18]. More direct chemical evidence has emerged from anaerobic incubations of sediment from Lake Constance[15] and of wetland soils in southeastern China[10] spiked with $^{14}CH_4$/$^{13}CH_4$ and $NO_2^-$ that led to the production of $^{14}CO_2$/$^{13}CO_2$, and from high resolution microprofiling of dissolved oxygen, $CH_4$, $NO_3^-$ and $NO_2^-$ with and without the addition of $NO_3^-$ in sediment cores of Lake Constance[16]. In the few cases where NC10 bacteria were detected in the water columns, albeit at low abundance, N-DAMO activity was not demonstrated. We present here results of incubation experiments with $^{15}N$-labelled $NO_3^-$ and $NO_2^-$ in the presence and absence of $CH_4$ along with a much larger data set, the first of its kind, on rates of denitrification, anammox and dissimilatory nitrate/nitrite reduction to ammonium (DNRA) from a number of dam-reservoirs in India. We also present results of a very large number of $N_2O$ measurements in the reservoirs, the first such report from any freshwater system in South Asia. The results of these measurements in conjunction with molecular data are used to gain insights into pathways of nitrogen loss from Indian freshwater reservoirs.

## Results

**Seasonal stratification and its impact on water chemistry.** Summer warming of surface waters resulted in strong stratification and consequent oxygen depletion in the hypolimnia of all 15 sampled reservoirs that are located over a wide latitudinal range (9.8°N−31.4°N−Supplementary Fig. 1, Supplementary Table 1) and differ in size, geology, climate and influence from human activities. In all cases, with the exception of two Himalayan reservoirs (Bhakra-Nangal and Tehri), and Rihand and Supa located in the Indo-Gangetic Plain and the Western Ghats,

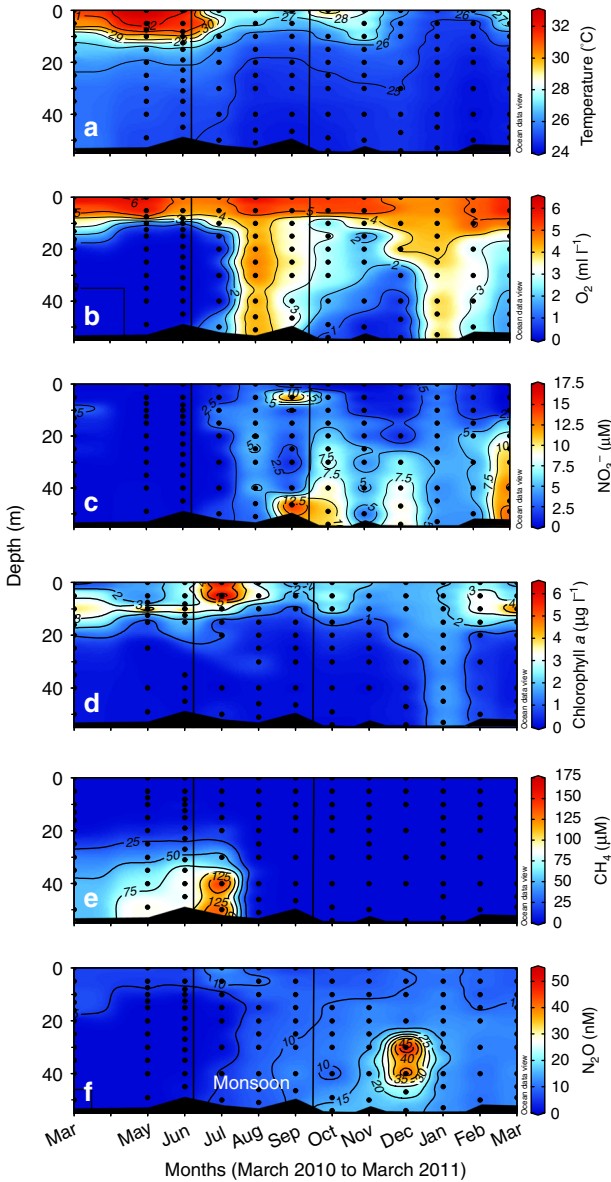

**Fig. 1** Annual cycles of physico chemical variables in Tillari Reservoir. Temperature (**a**), dissolved oxygen (**b**), nitrate (**c**), chlorophyll a (**d**), methane (**e**), and nitrous oxide (**f**) were measured once a month from March 2010 to March 2011. The summer monsoon period (June–September) is demarcated by black vertical lines

respectively, dissolved oxygen concentration fell below the detection limit (Supplementary Table 2). The latter two reservoirs were probably not sampled at the peak of oxygen depletion. The anoxic hypolimnia were characterised by low $NO_x^-$ ($NO_3^-$ + $NO_2^-$) concentrations (< 5 μM) in a majority of reservoirs and accumulation of $CH_4$ (Figs. 1, 2 and 3, Supplementary Fig. 2, Supplementary Table 2; also see ref. [19]). The highest $CH_4$ concentration measured in the present study was ~207 μM in Tillari Reservoir, further reinforcing the view that $CH_4$ accumulation in Indian dam-reservoirs is generally less than reported from several other tropical reservoirs[19]. Measurements of nitrogen to argon ratio ($N_2$/Ar) in four of these systems in summer (Supplementary Table 1) revealed excess $N_2$ in anoxic hypolimnia (up to 13.6 μM) seemingly produced from the reduction of $NO_3^-$ to $N_2$. These included the two most-frequently sampled reservoirs, Markandeya and Tillari (Supplementary Figs. 3 and 4), which experience

 

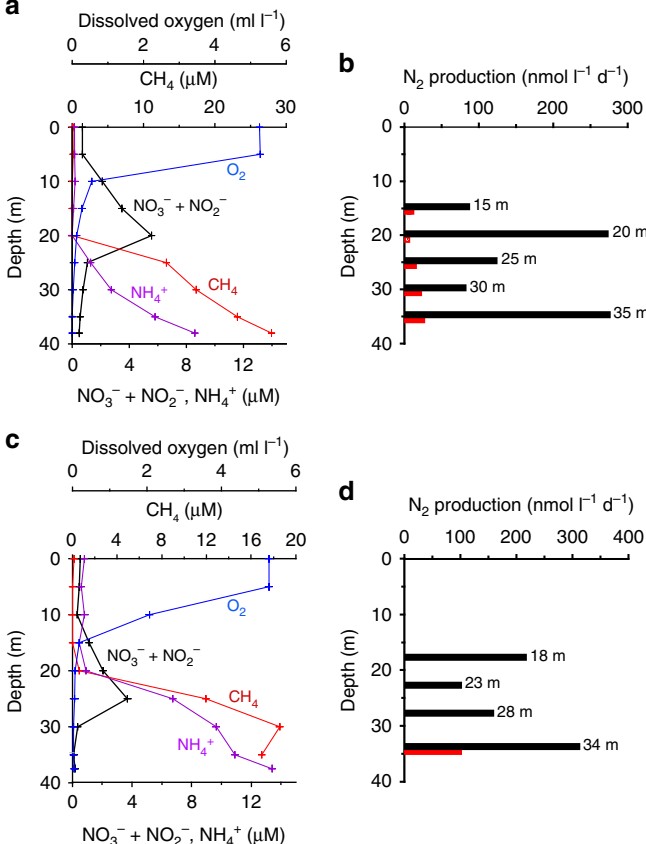

**Fig. 2** Water characteristics and denitrification rate in Tillari Reservoir. Depthwise variations in concentrations of dissolved oxygen, nitrate (+nitrite), ammonium and methane on 31.05.2011 (**a**), and 03.06.2011 (**c**). Total denitrification rate measured by isotope pairing experiments (black bars – with methane; red bars—without methane) on 31.05.2011 (**b**), and 03.06.2011 (**d**)

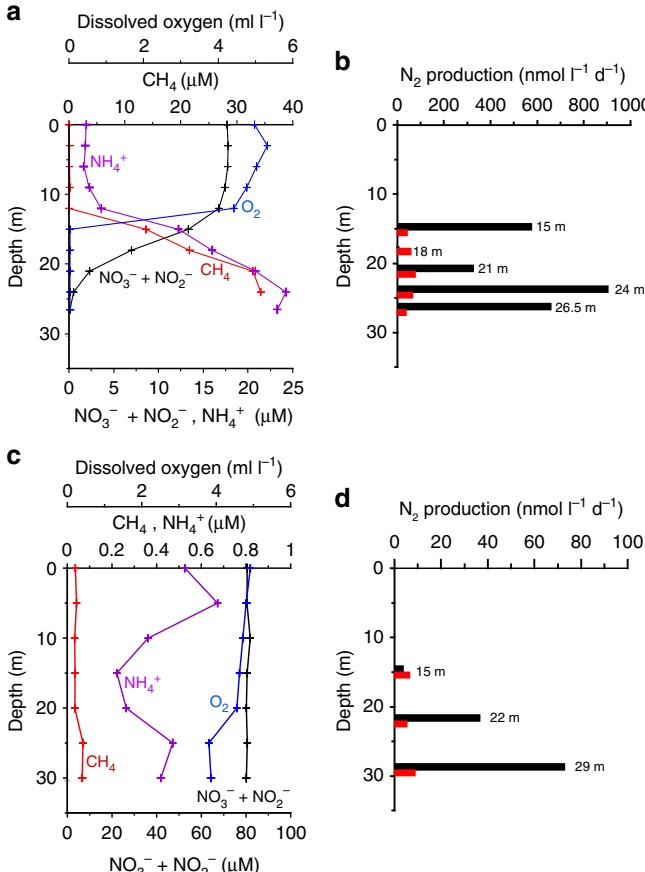

**Fig. 3** Water characteristics and denitrification rate in Markandeya Reservoir. Depthwise variations in concentrations of dissolved oxygen, nitrate (+nitrite), ammonium and methane on 15.06.2011 (**a**), and 12.01.2012 (**c**). Total denitrification rate measured by isotope pairing experiments (black bars – with methane; red bars – without methane) on 15.06.2011 (**b**), and 12.01.2012 (**d**)

contrasting degrees of anthropogenic impact. The Tillari Reservoir, located in relatively pristine foothills of the Western Ghats, was visited on an almost monthly basis from March 2010 to July 2014. The time-series for key parameters for the first 12 months are shown in Fig. 1. The next most frequently sampled reservoir—Markandeya Reservoir located in the Deccan Plateau—is much more affected by human activities (runoffs from agricultural fields and much denser population centres in the drainage basin of the Markandeya River). Although seasonal changes in water chemistry in this reservoir were similar to those observed in the Tillari Reservoir, the amplitude of variability was much larger. The maximum $NO_3^-$ concentration (~150 μM), for example, is about ten times higher than that recorded in the Tillari Reservoir. The reservoirs are dimictic—i.e., the water column is vertically mixed during winter and the southwest monsoon (June–September), when the reservoirs are recharged with water and nutrients (Fig. 1 and Supplementary Fig. 5). As expected, $NO_3^-$ regenerated from organic matter began to accumulate as oxygen levels declined in the hypolimnion after the establishment of stratification in early spring of 2011 in Tillari Reservoir (Fig. 1). The $NO_3^-$ concentration in the hypolimnion averaged around 12 μM on 10.03.2011. It decreased dramatically once the hypolimnion became anoxic, averaging around 2 μM on 31.05.2011 (data not shown). In 2010, however, when the first observation was made in Tillari Reservoir on 31 March, anoxia was already fully developed and the peak $NO_3^-$ concentration measured was only about 1 μM. This pattern of $NO_3^-$ buildup within the hypolimnion in late

winter-early spring and rapid loss in late spring-early summer was invariably observed in the following years (data not shown). Since $NO_3^-$ is often the dominant species of fixed nitrogen, the combined dissolved inorganic nitrogen (DIN = $NO_3^-$ + $NO_2^-$ + $NH_4^+$) concentration showed similar declining trend as $NO_3^-$ through the spring and summer (Supplementary Fig. 6 for Markandeya Reservoir). However, bottom-water $NH_4^+$ concentrations in late summer could occasionally be quite high (up to ~48 μM and 106 μM in Tillari and Markandeya, respectively) when the water turns mildly sulphidic.

A striking feature of nitrogen cycling in the reservoirs studied is the generally low concentrations of $NO_2^-$, as well as $N_2O$ in anoxic waters, unlike typical OMZs in marine systems. For instance, $NO_2^-$ in the Tillari Reservoir was below 0.1 μM in 327 out of 419 samples, exceeding 0.5 μM only in 2 samples with a maximum of 0.72 μM (Supplementary Fig. 7). Out of 815 measurements of $NO_2^-$ made in all reservoirs, only 27 values exceeded 0.5 μM (maximum 1.35 μM—Fig. 4). Similarly, in Tillari Reservoir low $N_2O$ concentrations (<10 nM) persisted throughout the summer in the presence of $CH_4$ at micromolar levels, rising in winter up to 53.1 nM when $CH_4$ content was three orders of magnitude lower (Fig. 1). Overall, $N_2O$ content was <20 nM in 681 samples (84% of all measurements, Fig. 4), and values in excess of 100 nM (maximum 357.2 nM) were recorded only in 34 samples (in 4% cases). None of the latter samples came from the epilimnion, and most were associated with $O_2 < 0.5$ ml l$^{-1}$ and relatively elevated $NO_2^-$ (Fig. 4). A majority of

these samples were from Markandeya (4 trips) with the remainder coming from Tillari (3 trips), Idukki and Koyna (1 trip each) over summer months. Thus, most of the reservoirs did not show significant $N_2O$ accumulation despite the prevalence of low-oxygen conditions.

**Anaerobic nitrogen transformation rates**. To identify the pathway(s) of nitrogen loss, we performed $^{15}N$-incubation experiments to determine rates of denitrification and anammox during periods when anoxic conditions prevailed in the following reservoirs: Koyna and Selaulim (once each); Idukki and Markandeya (three times each); and Tillari (five times) (Supplementary Table 3). Although the choice of these reservoirs was primarily dictated by logistic convenience, these reservoirs also differ considerably in terms of human impact. Surprisingly, only in one instance—Markandeya Reservoir in May 2007—was consistently high denitrification activity observed throughout the hypolimnion (Denitrification results in the production of $^{15}N^{15}N$, and also of $^{14}N^{15}N$ through isotope pairing of the $^{15}N$-labelled $NO_x^-$ with ambient non-labelled (i.e., $^{14}N$) $NO_3^-$ and/or $NO_2^-$). In this instance, $^{15}N^{15}N$ and $^{14}N^{15}N$ production rates from $^{15}NO_2^-$ incubations reached up to 832.1 and 111.4 nmol $N_2$ $l^{-1}$ $d^{-1}$, respectively (Supplementary Fig. 8), with the mean (±standard deviation) total denitrification rate (TDR)[20] at $1371 \pm 368$ nmol $N_2$ $l^{-1}$ $d^{-1}$ ($n = 5$). On other occasions, $^{15}N^{15}N$ and $^{14}N^{15}N$ production rates were generally low (Supplementary Table 4) with a mean TDR of $53.2 \pm 149.9$ nmol $N_2 l^{-1} d^{-1}$ (Table 1). Moreover, despite high $NH_4^+$ concentrations

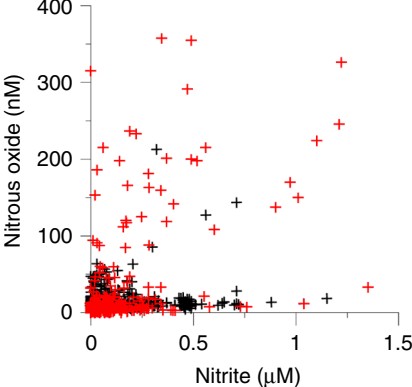

**Fig. 4** Nitrous oxide versus nitrite in all reservoirs. Red and black symbols denote samples with dissolved $O_2 \leq$ and $>0.5$ ml $l^{-1}$, respectively

(0.11–19.58 µM) in anoxic waters, the anammox rates determined from the production of $^{14}N^{15}N$ from $^{15}NH_4^+$ and $^{15}NO_2^-$ incubations were even lower (Supplementary Table 4), averaging $3.98 \pm 8.82$ nmol $N_2 l^{-1} d^{-1}$ and $12.62 \pm 22.47$ nmol $N_2 l^{-1} d^{-1}$ respectively (Table 1). The lower rates obtained from $^{15}NH_4^+$ incubations were likely due to the presence of ambient $^{14}NH_4^+$ in the samples. The low anammox activity was consistent with that detected via a combination of $^{15}NO_2^-$ and $^{14}NH_4^+$, thus ruling out low ambient $NO_2^-$ levels ($0.071 \pm 0.076$ µM; $n = 71$) being a major underlying reason here. The DNRA rates measured by production of labelled $NH_4^+$ from incubations involving $^{15}NO_3^-$ or $^{15}NO_2^-$ on nine trips to four reservoirs were also very low (Supplementary Table 4). DNRA was often not detectable, with the mean rates of $9.70 \pm 11.16$ and $8.71 \pm 13.44$ nmol $N l^{-1} d^{-1}$ obtained with $^{15}NO_3^-$ and $^{15}NO_2^-$ incubations, respectively (Table 1), being quite similar (Table 1). The only exception was Selaulim Reservoir, for which consistently high rates (70.97–162.14 nmol N $l^{-1} d^{-1}$ from $^{15}NO_2^-$) were measured throughout the anoxic hypolimnion during the one summer visit (Table 1).

**Enhancement of $N_2$ production by $CH_4$**. The measured rates of nitrogen loss in the methane-free $^{15}N$-incubations are much lower than expected from the in situ $NO_x^-$ and $N_2/Ar$ depth profiles. For example, in the Markandeya Reservoir sampled on 17.04.2015, the $NO_3^-$ concentration decreased from 53.9 µM from the top of the anoxic hypolimnion (12 m) to 27.5 µM close to the bottom (28 m); the excess $N_2$ within this layer ranged between 8.21 and 12.83 µM (Supplementary Fig. 4). The loss of $NO_3^-$ and accumulation of $N_2$ could only have occurred after the development of anoxic conditions, probably in less than a month, implying an average loss rate on the order of 1 µmol $NO_3^- l^{-1}$ d$^{-1}$. This is comparable to the average rate of nitrogen loss (~2 µmol $l^{-1} d^{-1}$) estimated by Deemer et al.[21] for Lakamas Lake, a small eutrophic reservoir in the State of Washington (USA), from the observed accumulation of $N_2$ in the hypolimnion in early summer. Intrigued by low rates of canonical denitrification, we incubated samples spiked with $^{15}NO_2^-$ and $^{15}NO_3^- + ^{14}NO_2^-$, with and without $CH_4$, on two trips to Tillari Reservoir and one trip to Markandeya Reservoir during periods of anoxia (in summer), and on one trip to Markandeya Reservoir in winter when the water column was well-oxygenated. The $^{15}NO_3^- + ^{14}NO_2^-$ amendment was designed to determine if $NO_3^-$ reduction to $NO_2^-$ was directly and efficiently coupled to the $N_2$ production process and its dependence on $CH_4$. The results of both the $^{15}NO_2^-$ and $^{15}NO_3^- + ^{14}NO_2^-$ incubations showed that $CH_4$ amendments significantly enhanced nitrogen loss

**Table 1 Rates of anaerobic nitrogen transformations**

| | Process rates (nmol $N_2$ $l^{-1} d^{-1}$) | | | Process rates (nmol N $l^{-1} d^{-1}$) | |
| --- | --- | --- | --- | --- | --- |
| | Denitrification from $^{15}NO_2^-$ | Anammox from $^{15}NO_2^-$ | Anammox from $^{15}NH_4^+$ | [b]DNRA from $^{15}NO_3^-$ | [c]DNRA from $^{15}NO_2^-$ |
| Range | N.D.–920.9[a] | N.D.–101.73 | N.D.–50.66 | N.D.–32.07 | N.D.–45.89 |
| Mean ± SD | 53.2 ± 149.9 | 12.62 ± 22.47 | 3.98 ± 8.82 | 9.70 ± 11.16 | 8.71 ± 13.44 |
| Median | 13.0 | 3.47 | 0.26 | 6.34 | 2.27 |
| N | 70[a] | 52 | 57 | 10 | 28 |
| Remarks on rates | >100 nM d$^{-1}$ = 6x | >100 nM d$^{-1}$ = 1x | >10 nM d$^{-1}$ = 7x | >10 nM d$^{-1}$ = 3x | >10 nM d$^{-1}$ = 8x |
| | >50 nM d$^{-1}$ = 14x | >50 nM d$^{-1}$ = 5x | >5 nM d$^{-1}$ = 12x | | |
| | <10 nM d$^{-1}$ = 30x | >10 nM d$^{-1}$ = 16x | N.D.=26x | | |
| | | N.D. = 14x | | | |

N.D. = not detectable
[a] Excluding 5 values (1024–1886, mean 1371 ± 368 nmol $N_2$ $l^{-1}$) from Markandeya on 3 May 2007 (See Supplementary Fig. 8)
[b] Excluding 4 values (2.94–118.45, mean 69.66 ± 50.01 nmol N $l^{-1}$) from Selaulim
[c] Excluding 5 samples (70.97–162.14, mean 116.70 ± 37.83 nmol N $l^{-1} d^{-1}$) from Selaulim

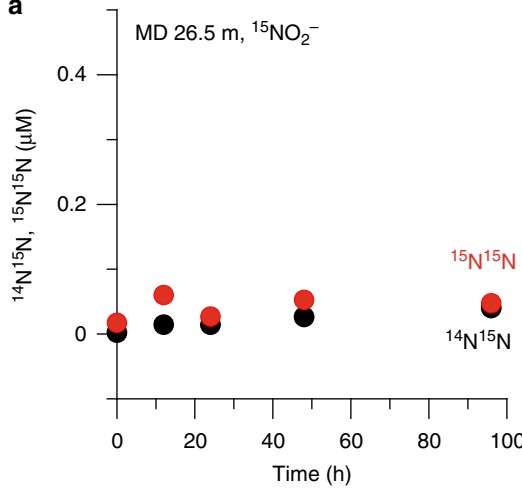

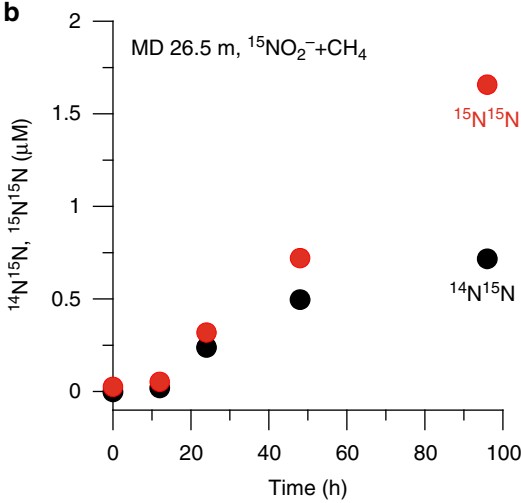

**Fig. 5** Stimulation of molecular nitrogen production by methane. Changes in concentrations of $^{14}N^{15}N$ and $^{15}N^{15}N$ versus time in a water sample collected from 26.5 m depth in Markandeya Reservoir on 15.06.2011 and incubated with $^{15}N$-labelled nitrite in the absence (**a**) and presence (**b**) of methane

through denitrification relative to experiments without $CH_4$ (Figs. 2 and 3, Supplementary Fig. 9). The nearly linear increase in $^{15}N^{15}N$ from $^{15}NO_2^-$ incubations (Fig. 5)—in all cases except one (Markandeya Reservoir, 18 m, 15.06.2011)—plus the linear increase in $^{15}N^{14}N$ from $^{15}NO_3^- + ^{14}NO_2^-$ incubations showed that $NO_2^-$ was the key oxidant in $N_2$ production, and that nitrate reduction was closely coupled to this process. Even when denitrification was detectable in the absence of $CH_4$ (>10 nmol $N_2$ $l^{-1} d^{-1}$; 10 out of 14 cases), denitrification rates increased by a factor of over 12 on average (mean 13.4; median 12.4) in $CH_4$-enriched samples. It should be noted that almost all of these samples originally contained significant amounts of $CH_4$ (up to ~34 μM), comparable to $CH_4$ introduced in the amendment experiments (~44 μM). However, measurements of $N_2$ production rates necessitate sparging of samples with helium before all incubation experiments in order to lower $N_2$ background prior to $^{15}N$-label additions, thus also removing dissolved $CH_4$ originally present in samples. Interestingly, during winter in the Markandeya Reservoir when the water column was well-oxygenated, potential $CH_4$-enhancement of $N_2$ production was also detected, albeit to a lesser extent (Fig. 3d). These samples were incubated under anoxic conditions with the dissolved oxygen removed

through helium-sparging. For these four trips to Tillari and Markandeya, DNRA rates measured from the same $^{15}NO_2^-$ incubations remained consistently low or undetectable (Supplementary Fig. 10), as did the anammox rates computed from these $^{15}NO_2^-$ incubations (0–12.24, mean $4.22 ± 4.52$ nmol $l^{-1} d^{-1}$). With such negligible anammox activity, exemplified by lack of significant $^{14}N^{15}N$ production in the absence of $CH_4$ in Markandeya Reservoir on 15.06.2011 (Fig. 5) despite high ammonium concentrations in bottom waters (Fig. 3a), the enhancement of $^{14}N^{15}N$ production in the presence of $CH_4$ must then be due to denitrification.

**Methanotrophic community structure**. Analyses of summer microbial community structure in the Markandeya Reservoir, based on amplicon sequencing of 16S rRNA genes, revealed the presence of the NC10 bacteria capable of N-DAMO in the hypolimnion, albeit at low relative abundance (0.003−0.022%). In comparison, the conventionally known aerobic methanotrophs were prevalent not only in the oxic but also in the anoxic layer, accounting for up to almost 14% of total community at the oxycline (12 m) (Fig. 6). Among the methanotrophs, all Type I, Type II, and Type III methanotrophs were present. Type I *Methylococcaceae* usually predominated (0.97–12.23%), of which *Methylocaldum* and *Methylomonas* were frequently identified though the majority remained unresolved at the genus level (Supplementary Table 5). Type II methanotrophs, including *Methylocystaceae* and *Methylobacteriaceae*, particularly the former, gained importance at the anoxic deeper depths, whereas Type III *Methylacidiphilales* superseded *Methylococcaceae* at the upper oxic depth (Supplementary Fig. 11). The presence of these methanotrophs, especially *Methylomonas*, *Methylobacter* and *Methylocaldum* were further corroborated by phylogenetic analyses of the biomarker gene *pmoA* that encodes particulate monooxygenase gene subunit A (Supplementary Fig. 12), though the primers used would have missed Type III methanotrophs as observed. The *pmoA* sequences directly related to NC10 bacteria were however not detected in samples collected in 2011 even using NC10-specific primers[8], most likely due to their low abundance. In addition, highly diverse community of potential denitrifiers were detected via phylogenetic analyses of biomarker gene encoding the $cd_1$-containing nitrite-reductase (NirS). Most sequences obtained were closely affiliated with sequences from methane-rich settings—rice-paddy soils and freshwater sediments. Notably, one of the Markandeya NirS cluster is associated with the Type I methanotroph *Methylomonas* sp. 16a (Supplementary Fig. 13). Though not as abundant as methanotrophs, organisms feeding on C-1 compounds known as methylotrophs also made a noticeable contribution (≤2.62%) at the same depths, especially *Methylophilaceae*. Moreover, small numbers of 16S rDNA sequences of methane-producing archaea were also retrieved especially at the lower anoxic depths based on 16S amplicon sequencing, with *Methanobacteriaceae, Methanocellaceae* and *Methanospirillaceae* being the most prominent (Supplementary Fig. 11, Supplementary Table 5). However, their abundances should not be treated as quantitative, as the primers used were meant for bacteria and not archaea.

## Discussion

With this much expanded database, our study lends strong support to the previously reported[19] absence of strong buildup of reactive nitrogen in Indian freshwater reservoirs, especially during summer when anoxic conditions develop in the hypolimnia of most reservoirs. The very low concentrations of $NO_x^-$ in the anoxic hypolimnia can only result from dissimilatory $NO_x^-$

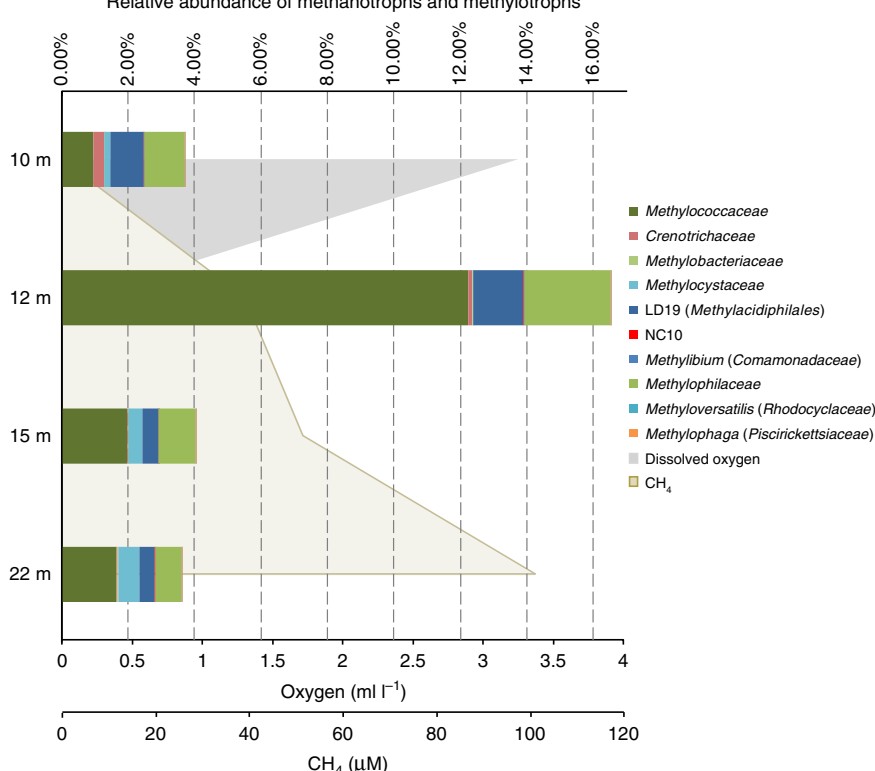

**Fig. 6** Relative percentage of known methanotrophs and methylotrophs based on amplicon sequencing of 16S rRNA genes in Markandeya Reservoir. Sampling was done on 17.06.2014. All methanotrophs (shown in bold) are at the family level except for the phylum NC10, while the methylotrophs are shown at genus level except for the family *Methylophilaceae*. Dissolved oxygen and methane ($CH_4$) levels at corresponding depths are shown as grey and light brown shaded areas, respectively. Dissolved oxygen was undetectable at or below 15 m

reduction to $N_2$ through denitrification and/or anammox, or to $NH_4^+$ through DNRA. Their occurrences would be consistent with the accumulation of both $N_2$ and $NH_4^+$ in the anoxic hypolimnia. Nonetheless, the measured rates of denitrification, anammox and DNRA in $CH_4$-free incubations—the first from any freshwater systems in South Asia—are surprisingly low (Table 1). In contrast, $N_2$-production rates are high when $CH_4$ is added to the $^{15}NO_x^-$ incubations. Most of this $N_2$-production can be attributed to denitrification, while anammox is only a minor nitrogen sink in the investigated Indian reservoirs. Our combined results provide the first direct evidence that $CH_4$, which is present throughout the anoxic hypolimnia, significantly enhances denitrification rates (over 12 times) in anoxic Indian lake waters.

It is highly unlikely that the above observations resulted from methodological artefacts, because the same method yielded consistently high denitrification rates in Markandeya Reservoir in May 2007 (Supplementary Fig. 8), as well as over the western continental shelf of India during seasonal anoxia in September 2011, where additional $CH_4$-enhanced $N_2$ production was not observed (Supplementary Fig. 14). Based on these results, we hypothesise that the stimulation of denitrification by $CH_4$ is restricted only to those aquatic environments where $CH_4$ builds up to sufficiently high levels to support the development of denitrifying methanotrophs. Unlike most marine systems, $CH_4$ can accumulate to very high levels (tens to hundreds of μM) in freshwater lakes and reservoirs, where methanogens face little competition from sulphate reducers for acetate[22,23]. This is consistent with the accumulation of $CH_4$ and the presence of methanotrophs observed in the hypolimnion during summer stratification period.

Consequently, diverse methanotrophic communities, both aerobic and anaerobic, benefitted from this carbon source. The denitrifying methanotrophs NC10 inhabited particularly in the lower anoxic depths (Fig. 6, Supplementary Fig. 11). Although they were not exactly abundant (≤0.023%), their low numbers are consistent with those reported in the few water-column studies to date, both marine[17,18] and freshwater[11]. Meanwhile, increasing evidence has demonstrated active occurrences of the traditionally considered aerobic methanotrophs and methylotrophs in hypoxic to anoxic conditions, including freshwater lakes, marine OMZs and anoxic fjords[24–28]. Even when NC10 bacteria were present alongside, these aerobic methanotrophs were usually at least an order of magnitude more abundant, most notably being the Type I *Methylococcaceae*, as has also been observed here in Markandeya Reservoir. In fact, some studies have even reported occurrences of *Methylobacter* (*Methylococcaceae*) only in anoxic but not oxic layers in some stratified lakes[27,28], indicating likely alternative anaerobic lifestyles.

Recent whole genome and environmental metagenome analyses have revealed that various assimilatory and dissimilatory nitrogen reduction genes, such as those encoding nitrate, nitrite, and nitric oxide- reductases, are relatively widespread among all Types I–III methanotrophs, possibly through horizontal gene transfers[25]. While denitrifying microbial communities are undeniably diverse and no primer sets to date have been adequate to capture the full spectrum of denitrifiers by targeting the biomarker gene *nirS*[29], our limited denitrifier screening has already revealed a potential relative to a denitrifying methanotroph *Methylomonas* sp., and *Methylomonas* was one of the more abundant genera found in Markandeya. Using cultured strains, nitrate reduction activity has been demonstrated with

*Methylomonas denitrificans*[30], while the Type II *Methylocystis* sp. SC2 has been shown capable of $N_2$ production from nitrate with methanol as a carbon source under anoxic conditions[31]. Genome analyses and anaerobic experiments with enrichment cultures indicate that the filamentous methanotroph *Crenothrix*, also present in Markandeya Reservoir (of *Crenotrichaceae*, Fig. 6, Supplementary Table 5), can reduce nitrate to $N_2O$[32], just like the methylotroph *Methylotenera*[33]. In other words, while the NC10 bacteria present were likely conducting N-DAMO, the much more prevalent conventional methanotrophs and methylotrophs could potentially be more important contributors to nitrogen reduction steps linked to methanotrophy.

It should be noted, however, that most of the genomes of conventional methanotrophs analysed to date still lack the nitrous oxide reductase gene (*nos*) responsible for the conversion of $N_2O$ to $N_2$, which seems to be at odds with our observed general lack of $N_2O$ accumulation in the reservoirs and $CH_4$-enhanced production of $^{15}N$-labelled $N_2$ in incubated samples. Nevertheless, the genetic capability to oxygenically dismutate nitric oxide (NO) to produce $N_2$ that bypasses $N_2O$—the pathway used by NC10 bacteria[5]—has recently been found to be more widespread in bacteria other than NC10: The *nod* gene encoding putative NO dismutase has been detected in the alkane-oxidising proteobacterium HdN1, and in a number of samples from contaminated aquifers and wastewater treatment systems[34]. A further search for this gene in public databases (NCBI and EMBL), including available whole genomes and environmental metagenomes, however, did not yield any reliable positive hits (below 36% amino acids identity) apart from those already reported[34], and none so far in conventional aerobic γ-/α-proteobacterial methanotrophs. Therefore, while environmental metagenomics studies of such systems have been sparse and the ability to dismutate NO remains to be fully explored within the methanotrophic community residing in the hypolimnia, our combined data thus far do not support N-DAMO by NC10 being the major contributor to our observed methane-induced denitrification.

It has been reported that ANME-2d archaea related to '*Candidatus* Methanoperedens nitroreducens' oxidise $CH_4$ with $NO_3^-$ under anaerobic conditions, but M. nitroreducens only reduces $NO_3^-$ to $NO_2^-$, which may then be converted to $N_2$ by either anammox or NC10 bacteria[35]. Despite the measured anammox rates being generally low and the abundance of anammox bacteria (≤0.013%) in the sequencing data being about half that of NC10 bacteria, it cannot be completely ruled out that the anammox bacteria were also involved in some $N_2$ production. Although several studies have reported co-occurrence of NC10 bacteria and ANME-2d archaea[36–38], the NC10 bacteria, or for that matter any other microbes mediating the production of $N_2$ from $NO_2^-$, do not have to rely only on M. nitroreducens for $NO_2^-$ supply, given multiple $NO_2^-$ sources, including the denitrifying methanotrophic bacteria. Significantly, in incubation experiments with $^{15}NO_3^-$ and $^{14}NO_2^-$ combined, we observed linear production of $^{15}N$-labelled $N_2$ (largely $^{14}N^{15}N$ due to high $^{14}NO_2^-$ levels) in the presence of $CH_4$ but not in its absence (Supplementary Fig. 9). This clearly shows the production of $NO_2^-$ from $NO_3^-$, which is rapidly converted to $N_2$, only in the presence of $CH_4$.

Hence, the $CH_4$-dependent production of $N_2$ from $NO_x^-$ is best explained by the concerted activity of a diverse microbial community comprising NC10 bacteria, canonical denitrifiers, anammox bacteria and aerobic methanotrophic α−and γ-proteobacteria. Considering the low abundance of NC10 bacteria, methanotrophic α−and γ-proteobacteria perhaps account for a large part of the observed effect. They could switch from respiring oxygen to nitrate, and might either produce $N_2$ themselves or N-compounds of intermediate redox states that are further reduced to $N_2$ by canonical denitrifiers or anammox

bacteria. Further assessment of single-cell activity and (meta) genomics/transcriptomics analyses are necessary to pinpoint the exact nature (single-organism versus syntrophic modes) of the observed denitrifying methanotrophy. The removal of $NO_3^-$ and $NO_2^-$ in the presence of $CH_4$ may explain the lack of large $NO_2^-$ accumulation observed in reservoirs as compared to several other marine (e.g., up to 23 μM for Peru[39], 16 μM off western India[40]) and freshwater (e.g., 18 μM in Lake Kinneret[41]) systems.

The data from Markandeya Reservoir on 03.05.2007 show that canonical denitrification could occasionally be an important nitrogen loss pathway. What determines the relative importance of canonical and $CH_4$-dependent denitrification is, however, unclear. Oxygen concentrations of a few 100's nanomolar, such as determined by the highly sensitive STOX (Switchable Trace amount Oxygen) sensors, can be sufficient to inhibit denitrification in seawater[42,43]. Thus, despite experimental evidence for aerobic denitrification in intertidal sediments[44], water-columns in regions like the Bay of Bengal and the Gulf of California do not exhibit pronounced nitrogen losses as dissolved oxygen, though vanishingly low, remains detectable within their OMZs[45–47]. Analogous to these observations, we propose that since the anoxic hypolimnia are located very close to the surface, canonical denitrification pathway may be hampered by frequent incursions of oxygen into the anoxic zones of these reservoirs. The sensitivity of denitrifying methanotrophs to oxygen at the very low range is poorly known, as the only study that assessed oxygen inhibition on NC10 bacteria utilised relatively high oxygen concentration (>2%)[48]. However, since these bacteria can themselves produce oxygen, they are expected to tolerate higher oxygen levels than canonical denitrifiers. Even though the latter are facultative anaerobes, they still require almost complete anoxia to switch from aerobic to nitrate/nitrite respiration[43]. The conventional methanotrophs, also postulated to be involved in our observed denitrifying methanotrophy, are known to be normally aerobes themselves. Therefore, given the availability of sufficient $CH_4$, denitrifying methanotrophs may out-compete canonical denitrifiers for $NO_2^-$. However, canonical denitrification is expected to become important if and when truly anoxic conditions persist long enough for the facultative bacteria to activate their denitrification enzymes, such as during calm weather conditions and strong stratification over extended periods, as presumably experienced during the May 2007 visit to Markandeya Reservoir.

In spite of the heavy anthropogenic impacts on Markandeya Reservoir, the nitrogen inventory therein decreased over the winter/spring mixing period when the water-column was well-oxygenated (from post-monsoon to winter to early spring; note that the only input of nitrogen to the reservoirs occurs during the monsoon). The reason for this decrease is unclear, but we speculate that it is due to the nitrogen uptake by phytoplankton followed by sedimentation of organic matter to the bottom. It is not known what fraction of nitrogen is permanently buried in the sediment, as opposed to how much is regenerated to $NH_4^+$ in the water column through aerobic respiration and how much is lost via benthic denitrification to $N_2/N_2O$. Benthic respiration is expected to be very important in shallow reservoirs[21]. It is likely that a significant amount of $NH_4^+$ in the water column originates from the sediment through degradation of organic matter via sulphate reduction. Smaller contribution could also come from DNRA both in the water column and sediments. The loss of $N_2$ may occur throughout the year through coupling between denitrification and methanotrophy within the sediment. This is supported by the appreciable $N_2$ production observed in the near-bottom samples of Markandeya in January 2012, when the oxygen-stripped samples were incubated with $^{15}NO_2^-$ and $CH_4$ (Fig. 3d).

Generally low nitrate concentrations in $CH_4$-bearing hypolimnia combined with reduced $CH_4$ concentrations in the absence of measurable oxygen are consistent with the proposed $CH_4$ involvement in nitrogen loss (Figs. 2a, c, 3a, Supplementary Fig. 2). This phenomenon may in part be responsible for the observed lower-than-expected[2] runoff of reactive nitrogen by rivers in South Asia[49], as well as the moderate build-up of $CH_4$ in the reservoirs[19]. Moreover, our data on $N_2O$ in Indian reservoirs, one of the largest of their kind from the freshwater systems in the world, indicate that a very small fraction of nitrogen loss is in the form of $N_2O$. As a by-product of nitrification and an intermediate of denitrification, $N_2O$ in oxygenated waters is mostly attributed to nitrification[50–52]. Although $N_2O$ production has been observed to be linearly related to oxygen consumption in some systems (e.g., in Lake Kizaki, Japan[50]), such a relationship is not universal[52]. Nitrification rates were not measured in the present study. However, going by the relationship reported from Lake Kizaki[50], a water sample having a temperature of 23 °C (typical of bottom waters in Markandeya Reservoir) is expected to contain ~74.4 nM $N_2O$ (8.3 nM saturation value plus 66.1 nM produced through nitrification) when dissolved oxygen has been fully consumed. Since ~84% of our measured $N_2O$ concentrations were below 20 nM, nitrification is most likely responsible for most of the $N_2O$ in the reservoirs sampled. The yield of $N_2O$ during denitrification in various environments is highly variable[53,54]. In the relatively well-studied lentic freshwater and marine ecosystems, it generally varies between 0.1 and 1.0%, although values as high as 6% have been reported[54]. Accordingly, $N_2O$ concentrations observed in anoxic freshwaters vary from below detection to several micromolar[21,52,55,56]. This is because while $N_2O$ is produced as well as consumed during denitrification, the highest (micromolar) $N_2O$ concentrations observed in both marine[22] and freshwater[52,55,56] ecosystems (a maximum of 88,400% $N_2O$ saturation in the anoxic hypolimnion of the Brookville Lake[56]) have been ascribed to this process. The moderately high peak $N_2O$ concentration measured in this study (357.2 nM, or 4,544% saturation) at 45 m in Tillari Reservoir on 03.05.2012, when the water column was very strongly stratified, is similarly attributed to $CH_4$-independent denitrification. This, however, was only a rare occurrence, and denitrification generally does not seem to be a major contributor to $N_2O$ production in the Indian freshwater reservoirs.

There exist striking similarities between conditions prevailing in the freshwater systems examined here and those found in semi-enclosed marine basins, such as the Black Sea, Baltic Sea, Cariaco Basin and the Saanich Inlet, where $CH_4$ accumulates in high concentrations in anoxic waters below the sill depth[22]. Concentrations of both $NO_2^-$ and $N_2O$ have been found to be consistently low above the oxic–anoxic interface (within the so called suboxic zone) in these basins as compared to the open ocean OMZs[22]. Hence, it is likely that the coupling between denitrification and methanotrophy may be more widespread and of much larger geochemical and environmental significance than realised thus far.

## Methods

**Sampling sites and measurements made.** Fifteen dam-reservoirs (see Supplementary Fig. 1 for locations) were chosen for sampling in the present study that extended over a period of 9 years (2006–2015). Of these reservoirs, some were visited only once during the summer season, but a few others were sampled repeatedly, covering different seasons (Supplementary Table 1). Observations made included a number of biogeochemical parameters like temperature, dissolved oxygen, $NO_3^-$, $NO_2^-$, $NH_4^+$, $N_2O$, $CH_4$ and $N_2/Ar$, as well as various nitrogen transformation rates determined by $^{15}N$-labelling experiments (Supplementary Tables 1 and 2). However, not all measurements could be made at the same time and in all reservoirs for logistic reasons. The two most frequently sampled reservoirs (Tillari and Markandeya) were chosen for more detailed studies because of logistic convenience, as they are within easy reach from the CSIR-National Institute

of Oceanography (NIO), Goa. They also represent two contrasting extents of anthropogenic impacts, the Tillari being more pristine.

**Sample collection and chemical analysis.** Sampling was carried out using an inflatable boat. Niskin samplers (5 litre) fitted with reversing thermometers and mounted on nylon ropes were used for the collection of water samples. Only a single sub-sample was taken from each depth for each measurement, including $^{15}N$-labelling experiments. Subsamples for dissolved oxygen were fixed immediately and analysed the same day following the Winkler procedure (precision ±0.03 ml l$^{-1}$). Subsamples for nutrients were stored in an ice box and frozen on arrival at the laboratory until analysis, usually carried out the next day following standard procedures[57] using a SKALAR analyzer. Separate subsamples were taken in ground-glass stoppered bottles from each sampling depth, one each for $N_2O$ and $CH_4$, and preserved with $HgCl_2$ (500 μl saturated solution/100 ml sample). The analyses were performed within a few days of collection by head-space extraction[58] with helium followed by injection into gas chromatographs equipped with electron capture detector (for $N_2O$) and flame ionisation detector (for $CH_4$) with precisions of ~4% and 7%, respectively. Samples for $N_2/Ar$ measurements were collected from four reservoirs in 60-ml serum bottles. Samples were preserved with 300 μl of saturated $HgCl_2$ solution and analysed three months later using a Hiden Quadrupole Membrane Inlet Mass Spectrometer. Preparation of standards and calibration were done following Hartnett and Seitzinger[59] and Charoenpong et al.[60]. Excess $N_2$ was computed from the $N_2/Ar$ ratio from $N_2$ solubility[61] at in-situ temperature.

**Labelling experiments.** For rate determination of denitrification, anammox and DNRA, $^{15}N$-labelling experiments were performed as per the procedures described in Holtappels et al.[62]. All sample processing and $^{15}N$-substrate additions were carried out at the sites, except for Tillari, samples from where were brought to NIO and processed on the same day. Briefly, 250 ml of sample was purged with ultra-pure helium for 10 min. Substrates were then added to get the following final concentrations: 8 μM $^{15}NO_2^-$ (denitrification and anammox), 8 μM $^{15}NH_4^+$ (anammox), 8 μM $^{15}NO_2^- + 8$ μM $^{14}NH_4^+$ (anammox + denitrification) and 16 or 20 μM $^{15}NO_3^-$ (DNRA). The sample was purged with helium for 5 more minutes and then dispensed into 12-ml Exetainer glass vials (Labco) by applying a slight helium overpressure and allowing sufficient overflow. Incubations were conducted at near in situ temperatures in the dark, and were terminated at predetermined time intervals by introducing a 2-ml helium headspace and adding 100 μl of saturated $HgCl_2$ solution. Inverted Exetainer vials were transferred as soon as possible to Max–Planck Institute for Marine Microbiology, Bremen, where $^{14}N^{15}N:^{14}N^{14}N$ and $^{15}N^{15}N:^{14}N^{14}N$ ratios in He-headspaces were determined by gas chromatography/isotope ratio mass spectrometry (Fisons VG Optima). The analysis was completed within a few weeks of sampling.

After analysis of labelled $N_2$ in the headspace, samples incubated with $^{15}NO_2^-$ and $^{15}NO_3^-$ were used to determine DNRA rates. $NH_4^+$ was oxidised to $N_2$ using sodium hypobromite (NaOBr) for measuring its isotopic composition. Briefly, 5 μM $^{14}NH_4^+$ and 200 μl of 4 M NaOBr were added to 5-ml of He-flushed sample in 6-ml Exetainers containing 1-ml He headspace. Isotopic ratios in $N_2$ in headspace were measured as described above.

For the experiments involving $CH_4$, deionised water filtered through 0.2 μm was saturated with $CH_4$. 10-ml of this water was added to samples (250 ml) previously sparged with helium and the incubations were carried out as for other samples after spiking with 20 μM $^{15}NO_2^-$ or 20 μM $^{14}NO_2^- + 20$ μM $^{15}NO_3^-$. Random analysis of $CH_4$-enriched samples ($n = 5$) yielded an average $CH_4$ concentration of ~44 μM, which is comparable to the observed in situ $CH_4$ concentrations in the samples (Figs. 2a, c, 3a).

**Detection of functional and 16S rRNA genes.** For the detection of biomarker functional genes, water samples were collected from Markandeya Reservoir on 15.06.2011 and filtered on to polycarbonate membrane filters or Sterivex cartridge filters (both of 0.22 μm pore size, Millipore). These filters were stored frozen until nucleic acids extractions. DNA was extracted from filter samples collected from 10 m, 12.5 m, 15 m, 17.5 m, 24 m, and a $CH_4$-amended incubation sample at 26.5 m. In addition, on a subsequent trip to the reservoir on 16.08.2011 when the water column became fully oxygenated and quite turbid following runoff during the monsoon, water sample was collected close to the bottom (30 m). This sample was filtered first through a 0.45 μm pore-size filter and then through a 0.2 μm filter, and analyses were performed on both filters. The *pmoA* and *nirS* genes were amplified with the primers A186-A689[63] and nirS1F-nirS6R[64]. The *pmoA* of NC10 bacteria were also specially targeted by both nested and non-nested PCR, as previously described[8], but no true *pmoA* sequences were yielded. PCR products of correct sizes, as checked with 2% agarose gel electrophoresis and visualised with SYBR Green staining, were gel-purified with the QIAquick PCR Purification Kit (Qiagen). Clone libraries were constructed for samples from 15 m, 26.5 m ($CH_4$-amended), 30 m (0.45 μm filtered) and 30 m (≥0.45 μm particulates) by using the TOPO TA Cloning Kit for Sequencing (pCR4 vector, Invitrogen). Ninety-six colonies were randomly picked from each clone library and PCR-screened with the primers M13F-M13R. Amplicons of 48 randomly chosen clone inserts were then submitted to GATC Biotech (Konstanz, Germany) for DNA sequencing. Phylogenetic

relationships were deduced based on amino acids sequences via neighbour-joining, maximum parsimony and maximum likelihood algorithms, with topologies further verified by bootstrapped resampling (×1000 for neighbour joining and ×100 for maximum parsimony and likelihood methods), by using the ARB software package[65].

Water samples from 0, 10, 12, 15, and 22 m depths (2 litre each) were collected again on 17.06.2014 from Markandeya Reservoir. These were filtered on site through 0.22 μm Sterivex cartridge filters (Millipore) using a peristaltic pump. The filters were preserved with DNA storage buffer until nucleic acid extraction. DNA was extracted from the filters using Mo Bio Power Water DNA isolation kit (Cat # 14900-50-NF) as per the manufacturer's protocol. From each of the samples, 1 μg of clean genomic DNA was used for further analysis. The bacterial community present in the sample was evaluated using Illumina MiSeq Platform by sequencing the V3 and V6 regions of the 16S rDNA (Genotypic Technology Pvt. Ltd., Bangalore, India). Briefly, the primers V3F (5′ CCAGACTCCTACGGGAGGCAG 3′ and V3R (5′CGTATTACCGCGGCTGCTG 3′) targeting V3 hypervariable regions and V6F (5′ TCGATGCAACGCGAAGAA 3′) and V6R (5′ ACATTTCACAACACGAGCTGACGA 3′) targeting V6 hypervariable regions of 16S rDNA genes were selected. Both forward and reverse primers were tagged with adaptor, pad and linker sequences. The PCR amplification was carried out and amplicons purified. The purified mixture was further processed as per MiSeq Reagent Kit Preparation Guide (Illumina, USA) and loaded on Illumina MiSeq for sequencing. After sequencing, the Illumina paired end raw reads was quality checked using SeqQC v2.1. The quality of demultiplexed raw read pairs was checked with FastQC (v 0.11.4; Babraham Bioinformatics). Forward and reverse reads with a maximum read length of 500 bp and a minimum Phred33 quality score of 0.8 were merged with the PANDAseq assembler software (v 2.8)[66]. The merged sequences were quality-checked again with the FastQC software. Open reference OTU clustering was subsequently performed under QIIME (MacQIIME v 1.9.1_20150604; Caporaso et al.[67]), using the UCLUST algorithm with a minimum sequence identity of 97% and the 16S rRNA Silva database (v 128)[68]. Singleton OTUs and sequences affiliated with chloroplastic rDNA were removed from the data set. Due to the short read lengths of V6 sequences that yield much fewer OTUs, only V3 results are presented here.

**Data availability**. The data used in the study will be archived at National Institute of Oceanography Data Centre (did.nio.org) and made available on request. Nucleotide sequences obtained from this study have been deposited in the Gen-Bank, under the accession numbers MG992020 to MG992185 for *nirS* genes, MG992186 to MG992311 for *pmoA* genes, and SRX3744137 to SRX3744145 for 16S rRNA genes.

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

## Acknowledgements

This research was supported by the European Commission through the award of a Marie Curie International Incoming Fellowship to SWAN to Max-Planck Institute of Marine Microbiology, Bremen, Germany; by the Max Planck Society (MPG); by the Council of Scientific & Industrial Research (CSIR, India) under the Network Project INDIAS IDEA; and by the Department of Science & Technology, Government of India, through the award of a J.C. Bose Fellowship to SWAN. CSIR also awarded Senior Research Fellowships to AS and GN. The authors are grateful to the authorities of various dams for providing them unhindered access to the reservoirs. This is NIO contribution number 6164.

## Author contributions

S.W.A.N., G.L., and M.M.M.K. designed the experiments. S.W.A.N., G.N., A.S., H.N., A.P., D.M.S., M.G., S.K., and S.D. participated in the field work. S.W.A.N., G.N., A.S., and H.N. performed $^{15}N$-labelling experiments. S.W.A.N., P.L., G.L., A.S., A.P., and S.D. analysed samples and processed data. M.D. assisted in processing of molecular data. S.W.A.N. and P.L. wrote the manuscript with input from all co-authors.
