## [Peer Review File(PDF 472 kb) · Nature Communications]

Reviewers' comments:

Reviewer #1 (Remarks to the Author):

Comments on the Ms. Methane stimulates massive nitrogen loss from freshwater reservoirs in India.

I have reviewed the paper entitled "Methane stimulates massive nitrogen loss from freshwater reservoirs in India" by Dr Naqvi and colleagues. The authors observed enhanced N₂ production in the presence of CH₄ in hypolimnia of freshwater reservoirs. Water column data, NO₃⁻ reduction processes and CH₄ amendment incubations are presented to support evidence on the coupling between denitrification and methanotrophy. Molecular analysis was also performed.

Overall, this is an interesting paper and a valuable addition to a novel research theme, however a few elements should be clearer. A large amount of data is presented but, as I understood it, results (from rates/amendments/molecular analysis) were not obtained concomitantly for each sample/site/survey. The range of experimental procedures was not done for the mentioned 8 reservoirs. Even for the more studied reservoir (Markandeya) the presented data were obtained with a gap of a few years: we have N₂ measurements from 04.2015; rates from 05.2007, 06.2011 and 01.2012; CH₄ amendments from 06.2011; and molecular analysis from 06.2011. What was done in each reservoir is scattered throughout the text but this made the reading and interpretation of results a bit tiresome. I think the manuscript would benefit with a more elaborated Methods section. Perhaps a table with surveys and experiments performed could be included.

Here are specific points to be addressed:

Introduction

Other field studies in water: Padilla et al. (2016), doi:10.1038/ismej.2015.262

Results

Pages 4-5: Was N₂/Ar measured in several reservoirs? In methods section you stated it was only measured in Markandeya in 04.2015.

Page 6: Table S1 is missing.

Methods

Please add sampling depths or depth intervals.

Were there analytical replicates for trace gases/N₂ measurements.

Were there incubation replicates for determination of processes?

Include how labelled NH₄⁺ was measured.

What are the major claims of the paper?

Enhanced production of N₂ linked to the presence of CH₄ in freshwater reservoirs in India.

Are the claims novel? If not, please identify the major papers that compromise novelty.

Coupled denitrification – methanotrophy has been previously reported but its environmental relevance is still unclear.

Will the paper be of interest to others in the field?

Yes.

Will the paper influence thinking in the field?

Yes.

Are the claims convincing? If not, what further evidence is needed?

Overall yes.

Are there other experiments that would strengthen the paper further? How much would they improve it, and how difficult are they likely to be?

A more systematic approach would have reinforced presented data but I understand that little can be done now.

Are the claims appropriately discussed in the context of previous literature?

Yes, although in the meantime I would recommend an additional look for recently published papers.

If the manuscript is unacceptable in its present form, does the study seem sufficiently promising that the authors should be encouraged to consider a resubmission in the future?

n/a

Is the manuscript clearly written? If not, how could it be made more accessible?

Yes.

Could the manuscript be shortened to aid communication of the most important findings?

The report on process rates is quite dense and, although important, the detailing is not essential to the major claims of the study.

Have the authors done themselves justice without overselling their claims?

Yes.

Have they been fair in their treatment of previous literature?

Yes.

Have they provided sufficient methodological detail that the experiments could be reproduced?

As stated above, I think that methodological detail could be improved.

Is the statistical analysis of the data sound?

n/a

Should the authors be asked to provide further data or methodological information to help others replicate their work? (Such data might include source code for modelling studies, detailed protocols or mathematical derivations).

Methods used are not new but additional detail would benefit the Ms.

Are there any special ethical concerns arising from the use of animals or human subjects?

n/a

Reviewer #2 (Remarks to the Author):

The paper ,Methane stimulates massive nitrogen loss from freshwater reservoirs in India ' by Naqvi et al. describes aspects of the nitrogen cycle in these man-made ecosystems using water column measurements, incubation experiments with stable isotopes and PCR-based detection of functional gene markers. Its major novel finding is the stimulation of denitrification by addition of dissolved methane, an observation not previously found for similar ecosystems. It remains, however, unclear which organisms and pathways are responsible for this activity. The authors also claim to present the largest dataset of nitrous oxide measurements in Indian freshwater bodies, but this is discussed only on the side. The data are relevant for microbiologists and limnologists (and possibly climate modellers, but this is too far from my expertise).

The paper was difficult to read, it certainly needs improvement in text structure and thorough proofreading by native English speakers. I have commented only on the most disturbing mistakes or unclear sentences.

Note to journal staff and authors: I apologize for the long time it took to finalize the review.

Some extra time was unfortunately necessary because it is very unhandy to comment on a version without line numbers.

My comments in detail:

Introduction (p.3&4)

The authors should be more critical to what they cite. Refs 7 & 11 (both from Scientific Reports) are very questionable papers. Ref. 15 is cited with a wrong title.

‘All field studies conducted so far have focussed on sediments with just one exception ...’ This is not correct. You cite yourself the paper of Hu et al (ref. 10), which focuses on paddy fields. Zhu et al. (2012) AEM 78: 8657-8665 is another, earlier example of a profound geochemical and qPCR profile, incubation & enrichment study (in a peatland). Both investigate soils, not sediments. I think you want to make the point that no previous study has focussed on water columns, which is correct as far as I know, but please be precise.

Results

I do not understand the first subtitle ‘Property distribution’. Please clarify.

P. 5, first paragraph: Why is it remarkable that nitrite concentrations are low in anoxic waters? Isn’t this to be expected if organic material is not the limiting factor? If you want to make such a point, please compare to literature values.

It does not become clear (here or in introduction), why Tillari and Markandeya reservoir were selected for more in-depth study, and generally the choice of reservoirs is not well explained.

P.6: This entire page is very difficult to read. Maybe you could partly replace the text by providing a (supplemental) table with an overview of rate measurements?

P.7 and Methods p. 14: 'spiked with 15NO_3^- (plus 14NO_2^-) and 15NO_2^- ' – this is confusing. Was 14NO_2^- added? What does it show in contrast to the experiments with 15NO_3^- and 15NO_2^- alone? What about background concentrations of nitrogenous compounds?

P.7 'It should be noted that almost all of these samples originally contained high CH_4 (up to $\sim 34 \mu\text{M}$).' This is not high; a lot of lakes reach saturation (around $1500 \mu\text{M}$) in anoxic bottom waters.

'Consequently, N_2 production rates are expected to be higher in

situ.' Why? Didn't you add methane to similar concentrations as in nature? So you hopefully mimic the natural conditions. Here and in other places the results are already discussed. This does make sense, but not if there is also a separate Discussion section. Please restructure.

P.8 The entire paragraph Molecular Analysis needs rewriting. Here are my main points of criticism:

'While Type II methanotrophs are generally regarded as

more stress-tolerant, those of Type I are usually found in methane-emitting environments with stimulated methanotrophic activity and growth upon nitrogen fertilization (18).' First, being stress-tolerant is not in contrast to inhabiting methane-emitting environments. Second, this is again discussion, not presentation of results, and third, this is a very questionable generalization. Even if true, it does not tell anything about the investigated ecosystem. Please correct and rephrase.

The next sentence 'Hence, the detection of particularly *Methylomonas*-, *Methylobacter* and *Methylocaldum*- like *pmoA* in Markandeya Reservoir likely reflects active methanotrophy therein.' also does not make sense. The detection of genes never tells anything more but the current presence (dead or alive, active or not, having thrived there or higher up in the water column) of a certain group of organisms.

'The *pmoA* sequences directly related to *M. oxyfera* were not detected in samples collected in 2011, but some of our retrieved *pmoA* sequences (e.g. Markandeya *pmoA11*, *pmoA18*) are in fact associated with the co-inhabitants of NC10 bacteria (Fig. S9).' Even for an insider this is very difficult to follow. From intensive study of Fig S9 and associated Genbank entries I think the authors mean that some of their sequences are closely related to *pmoA* sequences of Type I methanotrophs that have been found in the same enrichment culture as NC10 bacteria. But what does that tell, why is it worth mentioning? Also the next sentence is an overstatement; the presence of both methanotrophy and denitrification genes does not corroborate a close association between methanotrophy and denitrification, you will also find those genes in every garden soil or wastewater treatment plant together.

'The genetic capability to reduce nitrite to nitric oxide and/or nitrous oxide is known in several Type I and Type II methanotrophs (ref 19),' – Please discuss in context of more recent literature, eg. Kits et al 2015 (doi: 10.1111/1462-2920.12772 and doi: 10.3389/fmicb.2015.01072) and Padilla et al 2016 (doi: 10.3389/fmars.2017.00023).

'In any case, our detection of both nirS and pmoA in the same water samples supports our observed stimulation of denitrifying activity by methane.' See above; I don't believe there are many environments that do not contain both genes and the detection of genes is of very limited value.

You should also critically address the limitations of the primers you use for nirS, that were partly even known at time of publication (Braker 1998, ref. 39). Since then a vast amount of nirS have been found that are not covered by these primers (see for instance Heylen et al 2006 Env. Microb. DOI: 10.1111/j.1462-2920.2006.01081.x). Very likely, also the denitrifying methane-oxidizing Archaea have nir genes not amplified by these primers.

Discussion

P. 10 'By contrast, CH₄ can accumulate to very high levels (tens to hundreds of μM) in freshwater lakes and reservoirs due to a different production mechanism 22'. Very vague, what does different production mechanism mean? It is both methanogenesis. I think you want to state that there is less competition with sulfate-reducing bacteria for methanogenic substrates. Rephrase.

P. 11. Please repeat for clarity what was special in the 2007 incubations.

'It may be noted that the presence of oxygen may also affect functioning of denitrifying methanotrophs but perhaps at higher oxygen levels (ref 30). This may give them an advantage over canonical denitrifiers in shallow systems that are prone to frequent reoxygenation events.'

This does not make sense to me. First, ref. 30 used very high O₂ concentrations, which are not comparable to the concentrations measured in the deeper waters of the reservoirs. Second, nearly all 'canonical' denitrifiers are facultative aerobes, and consequently only their denitrification enzymes may be damaged, but not the organism as a whole. Additionally, some of the authors themselves have also shown denitrification in presence of oxygen (Gao et al 2010, Aerobic denitrification in permeable Wadden Sea sediments, doi:10.1038/ismej.2010.166).

P. 12 'It is not known what fraction of nitrogen is permanently buried in the sediment as opposed to how much is regenerated through respiration ...'. I guess you mean aerobic respiration (denitrification is also a type of respiration). Please correct.

Please also rephrase the following sentence 'It is likely that a significant amount of NH_4^+ in the water column originates from the sediment in addition to DNRA which, as described above, may be important in some cases'.

Please correct n-dano

'Moreover, coupling of anaerobic methane oxidation with NO_3^- reduction to NO_2^- by archaea and consumption of NO_2^- through anammox have also been reported(31).' It would be logical to first mention that NC10 bacteria and archaea have been found several times together (Raghoebarsing 2006 Nature 440: 918, Hu 2009 Env. Microb. Rep. 1: 377, Ettwig PNAS doi: 10.1073/pnas.1609534113)

Methods

P. 15 Please check language in last sentence.

Supplementary Material

Fig S7: I do not have a good alternative suggestion, but this is a difficult-to-grasp figure, because the several hundred cases on the right side are graphically not represented, only coded in the legend.

Fig. S9: Correct Verrumicribia - Verrucomicrobia

Fig. S11: Please use identical y-axes for both panels. 0 – 0.2 is sufficient

Reviewer #3 (Remarks to the Author):

Anaerobic microorganisms can couple the oxidation of methane to the reduction of sulfate, nitrate and nitrite, iron and manganese oxides. AOM is an important process to reduce the emission of the greenhouse gas methane. The distribution of anaerobic methanotrophs that can couple AOM to nitrate/nitrite reduction in natural environments and their contributions to the global methane and nitrogen cycles are hot research topics attracting a lot of attentions.

This manuscript provides evidences for the existence of methane driven denitrification process in several freshwater reservoirs in India. Overall, I believe their results are sound and important. However, I feel it is not significant enough to be published in Nat Com. Also due to the limitations of experimental plan and methods, some key evidences/results are missing, which reduces the strength of the paper. I will give more detailed comments below.

Significance

In recent years, there are many publications on the distribution of n-damo organisms in fresh water systems and their contributions to methane and nitrogen conversions. For details, see a mini review in EM Report: Nitrate- and nitrite-dependent anaerobic oxidation of methane.

The novelty of this study is that the authors focused on water column, while most of the previous studies targeted sediments. Since it is now an accepted fact that n-damo is a prevalent process in sediments of many different freshwater systems, we would expect it also happens in the anoxic water column above. Therefore, I think this is important results, but not significant enough for Nat Com. I would suggest the authors not only report the existence of methane driven denitrification in these dams, but also quantify and estimate the n-damo activity in these environments (it has been done for the wetland sediments), which will make this paper much more important.

Methodology

Some of the methods used by the authors are classic, e.g. isotopic N measurements. However there are many other aspects could have been better.

To date, there are only two confirmed n-damo microorganisms: *M. nitroreducens* and *M. oxyfera*. The authors tried to detect the key gene for *M. oxyfera*, but not for *M. nitroreducens*. For a paper reporting n-damo activity, this is a serious limitation.

No methane and/to CO₂ measurement results was shown. So the paper presents evidences for nitrogen conversion, however there is no conclusive evidence for methane conversion.

Overall, the authors showed that in samples that contains no or very low level of known anaerobic methanotrophs, the addition of methane can greatly enhance denitrification. More efforts should have been made to find out the responsible microorganisms.

No data for other important nutrients. We know AOM can couple to the reduction of sulfate, iron and manganese oxides. Also organic matters can drive denitrification. It is not essential but would be good if the authors can provide the information of these nutrients (in situ concentrations and variations during incubation tests).

Other comments

I also feel the manuscript is a bit sloppy. For example,

- Typos. n-dano in page 12, computed in Fig S2 caption

- Sloppy sentences. e.g...More direct chemical evidence has emerged from anaerobic incubations of sediment from Lake Constance¹⁴ and of wetland soils in southeastern China¹⁰ spiked with ¹⁴CH₄ and NO₂⁻ that led to production of ¹⁴CO₂...in reference 10 paper, they used ¹³CH₄, not ¹⁴CH₄.

- Strange numbers. 53.2±149.9 nmol, 0.071±0.076 μM; not sure you can get negative values there.

- Wrong reference styles. e.g. ...found to be a more important pathway to N₂ (refs. 25-29)...A The pmoA and nirS genes were amplified with the primers A186-A689 (ref. 39) and nirS1F-nirS6R (ref. 40).

- Use shorthand without defining the full term first, e.g. NIO?? OMZ.

· Wrong figure caption or legends, e.g. Fig S2, legend should be NO_3+NO_2 ? Fig S6, caption suggest ammonium data will be presented, but not in the figure.

REVIEWER # 1

Reviewer's Comment

I have reviewed the paper entitled “Methane stimulates massive nitrogen loss from freshwater reservoirs in India” by Dr Naqvi and colleagues. The authors observed enhanced N_2 production in the presence of CH_4 in hypolimnia of freshwater reservoirs. Water column data, NO_3^- reduction processes and CH_4 amendment incubations are presented to support evidence on the coupling between denitrification and methanotrophy. Molecular analysis was also

performed. Overall, this is an interesting paper and a valuable addition to a novel research theme, however a few elements should be clearer.

Authors' response

We thank the Reviewer for her/his encouraging and very thoughtful review. We have tried to do our best to address the issues raised by her/him as described below.

I(a). Reviewer's Comment

A large amount of data is presented but, as I understood it, results (from rates/amendments/molecular analysis) were not obtained concomitantly for each sample/site/survey. The range of experimental procedures was not done for the mentioned 8 reservoirs. Even for the more studied reservoir (Markandeya) the presented data were obtained with a gap of a few years: we have N₂ measurements from 04.2015; rates from 05.2007, 06.2011 and 01.2012; CH₄ amendments from 06.2011; and molecular analysis from 06.2011. What was done in each reservoir is scattered throughout the text but this made the reading and interpretation of results a bit tiresome. I think the manuscript would benefit with a more elaborated Methods section. Perhaps a table with surveys and experiments performed could be included.

Authors' Response

We have expanded the Methods section substantially and included two tables (Supplementary Tables 1 and 2) providing details of dates and measurements made.

I(b). Reviewer's Comment

Introduction

Other field studies in water: Padilla et al. (2016), doi:10.1038/ismej.2015.262

Authors' Response

In addition to the reference suggested by the Reviewer (ref. 17, line 496), we have also included additional references of recent work (refs 18, line 498; ref. 24, line 510).

I(c). Reviewer's Comment

Results

Pages 4-5: Was N₂/Ar measured in several reservoirs? In methods section you stated it was only measured in Markandeya in 04.2015.

Authors' Response

N₂/Ar was measured in 4 reservoirs. Details have been provided in Supplementary Table 1. We have also plotted data from Tillari Reservoir in Supplementary Fig. 3.

I(d). Reviewer's Comment

Page 6: Table S1 is missing.

Authors' Response

The new Supplementary Tables 1-3 provide more details than what Table S1 contained.

I(e). Reviewer's Comment

Methods

Please add sampling depths or depth intervals.

Authors' Response

Depths of samplings are provided in Supplementary Table 2 and also in the figures.

I(f). Reviewer's Comments

Were there analytical replicates for trace gases/N₂ measurements.

Were there incubation replicates for determination of processes?

Authors' Response

No. The measurements including tracer experiments were made on single samples (now clarified in line 344-345).

I(g). Reviewer's Comments

Include how labelled NH₄⁺ was measured.

Authors' Response

Included in lines 375-378).

I(h). Reviewer's Comments

Are the claims appropriately discussed in the context of previous literature?

Yes, although in the meantime I would recommend an additional look for recently published papers.

Authors' Response

We have included several additional references of recent work (ref 14, line 487; ref 17, line 496; ref 18; line 498; ref 23, line 507; ref 24, line 510; ref 29, line 523; ref 32, line 532) .

I(i). Reviewer's Comments

Could the manuscript be shortened to aid communication of the most important findings?

The report on process rates is quite dense and, although important, the detailing is not essential to the major claims of the study.

Authors' Response

We submit that the rates of denitrification, anammox and DNRA, being reported for the first time from any aquatic (freshwater) environment in South Asia, are very important because they are inconsistent with the observed lack of reactive nitrogen accumulation in a region that consumes about one-fifth of the total synthetic nitrogen used globally. Thus, these results indicate a hitherto unrecognized mode of nitrogen loss. We show that the nitrogen loss is driven by methane. However, we accept that the earlier discussion was "dense". We have tried to make it more easily digestible by simplifying and reducing the text and including a table (Table 1).

1(j). Reviewer's Comments

Have they provided sufficient methodological detail that the experiments could be reproduced?

As stated above, I think that methodological detail could be improved.

Should the authors be asked to provide further data or methodological information to help others replicate their work? (Such data might include source code for modelling studies, detailed protocols or mathematical derivations).

Methods used are not new but additional detail would benefit the Ms.

Authors' Response

We have now expanded the Methods section substantially as suggested by the Reviewer.

REVIEWER # 2

Reviewer's Comments

The paper ,Methane stimulates massive nitrogen loss from freshwater reservoirs in India ' by Naqvi et al. describes aspects of the nitrogen cycle in these man-made ecosystems using water column measurements, incubation experiments with stable isotopes and PCR-based detection of functional gene markers. Its major novel finding is the stimulation of denitrification by addition of dissolved methane, an observation not previously found for similar ecosystems.

Authors' Response

We thank the Reviewer for her/his encouraging and very thoughtful review. We have tried to do our best to address the issues raised by her/him as detailed below.

2(a). Reviewer's Comments

It remains, however, unclear which organisms and pathways are responsible for this activity.

Authors' Response

In the revision we have attempted to address the Reviewer's concern. Based on the additional data and discussion added to the manuscript we conclude that the CH₄-dependent production of N₂ from NO₂⁻ is best explained by a diverse denitrifying methanotroph community comprising both NC10 bacteria and conventionally 'aerobic' methanotrophs – the latter of which can either switch to denitrifying mode themselves, or work in syntrophy with other microbes with denitrifying ability including oxygenic nitric oxide dismutation. However, further assessment of single-cell activity and (meta)genomics /transcriptomics analyses are necessary to pinpoint exact nature (single-organism versus syntrophic modes) of the observed denitrifying methanotrophy.

2(b). Reviewer's Comments

The authors also claim to present the largest dataset of nitrous oxide measurements in Indian freshwater bodies, but this is discussed only on the side. The data are relevant for microbiologists and limnologists (and possibly climate modellers, but this is too far from my expertise).

Authors' Response

The N₂O data set we are presenting is not only the first from any freshwater ecosystem in South Asia, but also the largest from any data set on N₂O in freshwater in the world, as far as we know. The lack of large N₂O accumulation is indeed intriguing, given the huge nitrogen loading expected from human activities. Our purpose of including these data here is to make the point that the generally low N₂O concentrations (only 4% of the values exceeded 100 nM and in a majority of reservoirs the highest concentration was below 25 nM) indicate unusual nitrogen cycling.

2(c). Reviewer's Comments

The paper was difficult to read, it certainly needs improvement in text structure and thorough proofreading by native English speakers. I have commented only on the most disturbing mistakes or unclear sentences.

Authors' Response

We have strived to make the manuscript easier to read. We have restructured it by clearly separating Results and Discussion and bringing the sub-section on N₂O under the first sub-section now titled "Seasonal stratification and its impact on water chemistry". The Discussion part has also been thoroughly revised.

2(d). Reviewer's Comments

Some extra time was unfortunately necessary because it is very unhandy to comment on a version without line numbers.

Authors' Response

We apologize. The revised text includes line numbers.

2(e). Reviewer's Comments

Introduction (p.3&4)

The authors should be more critical to what they cite. Refs 7 & 11 (both from Scientific Reports) are very questionable papers. Ref. 15 is cited with a wrong title.

Authors' Response

We have dropped Reference 7, but have retained Reference 11 as this is the only published report - in a coveted journal - on NC10 bacteria in the water column of a dam-reservoir. The inclusion of latter does not in any way affect our conclusions. The title of the paper by Deutzmann et al. has been corrected (ref 16, line 493).

2(f). Reviewer's Comments

‘All field studies conducted so far have focussed on sediments with just one exception ...’

This is not correct. You cite yourself the paper of Hu et al (ref. 10), which focuses on paddy fields. Zhu et al. (2012) AEM 78: 8657-8665 is another, earlier example of a profound geochemical and qPCR profile, incubation & enrichment study (in a peatland). Both investigate soils, not sediments. I think you want to make the point that no previous study has focussed on water columns, which is correct as far as I know, but please be precise.

Authors' Response

We have made suitable changes following the Reviewer's advice (Lines 36-44).

2(g). Reviewer's Comments

Results

I do not understand the first subtitle 'Property distribution'. Please clarify.

Authors' Response

The subtitle has been changed to "Seasonal stratification and its impact on water chemistry" (starting line 55) and the previously subsection on N₂O distribution has also been merged with this subsection.

2(h). Reviewer's Comments

P. 5, first paragraph: Why is it remarkable that nitrite concentrations are low in anoxic waters? Isn't this to be expected if organic material is not the limiting factor? If you want to make such a point, please compare to literature values.

Authors' Response

A large number of oxygen-depleted environments, especially in the open ocean and continental shelves, are characterized by the accumulation of NO₂⁻, an intermediate of denitrification. In fact the "secondary nitrite maximum" is invariably associated with the most intense nitrogen loss in the ocean. NO₂⁻ accumulation has also been observed in freshwater systems. In India, hundreds of anoxic groundwater samples we have collected so far invariably contained NO₂⁻ in concentrations of tens of micromol/liter. By contrast, NO₂⁻ concentrations in anoxic hypolimnia are often close to the detection limit and in only 27 cases out of 815 measurements made by us nitrite concentration exceeded 0.5 μM (maximum 1.35 μM). While it is not entirely clear why nitrite accumulates in many denitrifying systems, this is apparently not due to lack of organic substrate because the highest concentrations occur in the most productive systems. We believe the low nitrite concentrations being reported here are significant and intriguing as they are probably related to high CH₄, which is oxidized by microbes using NO₂⁻. We have added three additional references (37-39, lines 543-549) and a brief discussion in response to the Reviewer's comment (lines 271-274).

2(i). Reviewer's Comments

It does not become clear (here or in introduction), why Tillari and Markandeya reservoir were selected for more in-depth study, and generally the choice of reservoirs is not well explained.

Authors' Response

The choice was largely based on logistical convenience - they both are easily reachable from Goa. They also happen to experience different degrees of human impact. We have stated this in the revision (Lines 108-110, 337-341).

2(j). Reviewer's Comments

P.6: This entire page is very difficult to read. Maybe you could partly replace the text by providing a (supplemental) table with an overview of rate measurements?

Authors' Response

We have made suitable changes following the referee's advice. Please see our response to Comment 1(i) for Reviewer #1.

2(k). Reviewer's Comments

P.7 and Methods p. 14: 'spiked with $^{15}\text{NO}_3^-$ (plus $^{14}\text{NO}_2^-$) and $^{15}\text{NO}_2^-$ ' – this is confusing. Was $^{14}\text{NO}_2^-$ added? What does it show in contrast to the experiments with $^{15}\text{NO}_3^-$ and $^{15}\text{NO}_2^-$ alone? What about background concentrations of nitrogenous compounds?

Authors' Response

The ambient nitrite concentrations were very low, as already discussed above. $^{14}\text{NO}_2^-$ was added along with $^{15}\text{NO}_3^-$ to see if $^{15}\text{NO}_2^-$ produced from the reduction of $^{15}\text{NO}_3^-$ was combining with $^{14}\text{NO}_2^-$ to produce $^{15}\text{N}^{14}\text{N}$. As shown in Supplementary Fig. 13, this happens only when CH_4 is present (Clarified in Lines 130-135).

2(l). Reviewer's Comments

P.7 'It should be noted that almost all of these samples originally contained high CH_4 (up to $\sim 34 \mu\text{M}$).' This is not high; a lot of lakes reach saturation (around $1500 \mu\text{M}$) in anoxic bottom waters. 'Consequently, N_2 production rates are expected to be higher in situ.' Why? Didn't you add methane to similar concentrations as in nature? So you hopefully mimick the natural conditions.

Authors' Response

We agree that CH_4 accumulation in Indian reservoirs is substantially less than in several other reservoirs (which itself is a significant finding that probably reflects greater (anaerobic) methane oxidation). However, the observed concentrations are still three to four orders of magnitude higher than those observed in oxygenated waters and even in many anoxic waters (including the Indian shelf). The point we were trying to make - and apparently did not do a good job in the first instance - is that the samples originally contained CH_4 that would facilitate N_2 production in the natural environment. The dissolved CH_4 was purged out due to pre-incubation sparging with helium and so the controls (which were CH_4 -free) exhibited little production of labelled N_2 . We have now modified the relevant text to make this clearer (lines 141-146).

2(m). Reviewer's Comments

Here and in other places the results are already discussed. This does make sense, but not if there is also a separate Discussion section. Please restructure.

Authors' Response

This is an excellent suggestion. We have followed the Reviewer's advice. The restructured manuscript now does not have much discussion under the "Results" section.

2(n). Reviewer's Comments

P.8 The entire paragraph Molecular Analysis needs rewriting. Here are my main points of criticism:

'While Type II methanotrophs are generally regarded as more stress-tolerant, those of Type I are usually found in methane-emitting environments with stimulated methanotrophic activity and growth upon nitrogen fertilization (18).' First, being stress-tolerant is not in contrast to inhabiting methane-emitting environments. Second, this is again discussion, not presentation of results, and third, this is a very questionable generalization. Even if true, it does not tell anything about the investigated ecosystem. Please correct and rephrase.

The next sentence 'Hence, the detection of particularly *Methylomonas*-, *Methylobacter* and *Methylocaldum*- like *pmoA* in Markandeya Reservoir likely reflects active methanotrophy

therein.’ also does not make sense. The detection of genes never tells anything more but the current presence (dead or alive, active or not, having thrived there or higher up in the water column) of a certain group of organisms.

‘The pmoA sequences directly related to *M. oxyfera* were not detected in samples collected in 2011, but some of our retrieved pmoA sequences (e.g. Markandeya pmoA11, pmoA18) are in fact associated with the co-inhabitants of NC10 bacteria (Fig. S9).’ Even for an insider this is very difficult to follow. From intensive study of Fig S9 and associated Genbank entries I think the authors mean that some of their sequences are closely related to pmoA sequences of Type I methanotrophs that have been found in the same enrichment culture as NC10 bacteria. But what does that tell, why is it worth mentioning? Also the next sentence is an overstatement; the presence of both methanotrophy and denitrification genes does not corroborate a close association between methanotrophy and denitrification, you will also find those genes in every garden soil or wastewater treatment plant together.

‘The genetic capability to reduce nitrite to nitric oxide and/or nitrous oxide is known in several Type I and Type II methanotrophs (ref 19),’ – Please discuss in context of more recent literature, eg. Kits et al 2015 (doi: 10.1111/1462-2920.12772 and doi: 10.3389/fmicb.2015.01072) and Padilla et al 2016 (doi: 10.3389/fmars.2017.00023).

‘In any case, our detection of both nirS and pmoA in the same water samples supports our observed stimulation of denitrifying activity by methane.’ See above; I don’t believe there are many environments that do not contain both genes and the detection of genes is of very limited value.

You should also critically address the limitations of the primers you use for nirS, that were partly even known at time of publication (Braker 1998, ref. 39). Since then a vast amount of nirS have been found that are not covered by these primers (see for instance Heylen et al 2006 *Env. Microb.* DOI: 10.1111/j.1462-2920.2006.01081.x). Very likely, also the denitrifying methane-oxidizing Archaea have nir genes not amplified by these primers.

Authors' Response

This entire section has been rewritten taking into account the Reviewer's comments. We now focus on the new 16S rRNA amplicon sequencing data (instead of functional gene dataset), which gave us an overview of the microbial community structure but with the focus on methanotrophic microbial communities. The dataset - though briefly mentioned before, has been reanalysed against the latest ARB-SILVA data set instead of previously Greengenes, which yielded a lot more robust classification, and also semi-quantitative data to compare community structure at various depths examined. These data are presented in Fig 5 and Supplementary Table 4. Consequently, the criticism from Reviewer on the pmoA and nirS interpretation does not apply any more. Although we still show the pmoA and nirS data in the supplement, they only serve as additional support to the discussion based on 16S data.

More recent publications have been included in the references, including all those suggested by the reviewer.

There are also comments on the limitations the chosen primers in Results section.

2(o). Reviewer's Comments

Discussion

P. 10 'By contrast, CH₄ can accumulate to very high levels (tens to hundreds of μM) in freshwater lakes and reservoirs due to a different production mechanism 22'. Very vague, what does different production mechanism mean? It is both methanogenesis. I think you want to state that there is less competition with sulfate-reducing bacteria for methanogenic substrates. Rephrase.

Authors' Response

Thanks. Yes, that is what we meant. We have made suitable changes in the text following the Reviewer's advice (Lines 204-208).

2(p). Reviewer's Comments P. 11. Please repeat for clarity what was special in the 2007 incubations.

Authors' Response

We have made suitable changes in the text following the Reviewer's advice (Lines 295-299).

2(q). Reviewer's Comments

'It may be noted that the presence of oxygen may also affect functioning of denitrifying methanotrophs but perhaps at higher oxygen levels (ref 30). This may give them an advantage over canonical denitrifiers in shallow systems that are prone to frequent reoxygenation events.'

This does not make sense to me. First, ref. 30 used very high O₂ concentrations, which are not comparable to the concentrations measured in the deeper waters of the reservoirs. Second, nearly all 'canonical' denitrifiers are facultative aerobes, and consequently only their denitrification enzymes may be damaged, but not the organism as a whole. Additionally, some of the authors themselves have also shown denitrification in presence of oxygen (Gao et al 2010, Aerobic denitrification in permeable Wadden Sea sediments, doi:10.1038/ismej.2010.166).

Authors' Response

We are, of course, aware of the Gao et al. (2010). However, we point out that there is overwhelming evidence that oxygen inhibits N-loss in water column when present even in traces (sub-micromolar concentrations) in areas such as the Bay of Bengal and Gulf of California. And while many of the denitrifiers may be facultative and genetically equipped to carry out denitrification the denitrifying genes do not seem to be expressed in the presence of traces of oxygen. What we are proposing is that low denitrification activity in the Indian reservoirs in the absence of CH₄ could be because the O₂ concentrations are often kept above that threshold (functional anoxia) as in many oceanic oxygen minimum zones. By contrast denitrifying heterotrophs that are actually postulated to produce O₂ may have a higher tolerance for O₂. However, in view of the Reviewer's criticism we have rephrased the text appropriately (Lines 278-286).

2(r). Reviewer's Comments

P. 12 'It is not known what fraction of nitrogen is permanently buried in the sediment as opposed to how much is regenerated through respiration ...'. I guess you mean aerobic respiration (denitrification is also a type of respiration). Please correct.

Please also rephrase the following sentence 'It is likely that a significant amount of NH₄⁺ in the water column originates from the sediment in addition to DNRA which, as described

above, may be important in some cases’.

Authors' Response

We have made suitable changes following the Reviewer's advice (Lines 305-311).

2(s). Reviewer's Comments

Please correct n-dano

Authors' Response

Done.

2(t). Reviewer's Comments

‘Moreover, coupling of anaerobic methane oxidation with NO_3^- reduction to NO_2^- by archaea and consumption of NO_2^- through anammox have also been reported (31).’ It would be logical to first mention that NC10 bacteria and archaea have been found several times together (Raghoebarsing 2006 Nature 440: 918, Hu 2009 Env. Microb. Rep. 1: 377, Ettwig PNAS doi: 10.1073/pnas.1609534113)

Authors' Response

We have made suitable changes following the Reviewer's advice (Lines 257-258; refs 34-36, Lines 537-542).

2(u). Reviewer's Comments

Methods

P. 15 Please check language in last sentence.

Authors' Response

Done.

2(v). Reviewer's Comments

Supplementary Material

Fig S7: I do not have a good alternative suggestion, but this is a difficult-to-grasp figure, because the several hundred cases on the right side are graphically not represented, only coded in the legend.

Authors' Response

We have removed the Figure and included a Table (Supplementary Table 4) listing the same data.

2(w). Reviewer's Comments

Fig. S9: Correct Verrumicribia - Verrucomicrobia

Authors' Response

Done (the figure is now renumbered as Supplementary Figure 10).

REVIEW # 3

Reviewer's Comments

This manuscript provides evidences for the existence of methane driven denitrification process in several freshwater reservoirs in India. Overall, I believe their results are sound and important. However, I feel it is not significant enough to be published in Nat Com. Also due to the limitations of experimental plan and methods, some key evidences/results are missing, which reduces the strength of the paper. I will give more detailed comments below.

Authors' Response

We thank the Reviewer for her/his encouraging and in-depth review. We also thank her/him for pointing out the errors that had unfortunately crept into the initial submission. We have tried to do our best to address the issues raised by her/him.

3(a). Reviewer's Comments

The novelty of this study is that the authors focused on water column, while most of the previous studies targeted sediments. Since it is now an accepted fact that n-damo is a prevalent process in sediments of many different freshwater systems, we would expect it also happens in the anoxic water column above. Therefore, I think this is important results, but not significant enough for Nat Com. I would suggest the authors not only report the existence of methane driven denitrification in these dams, but also quantify and estimate the n-damo activity in these environments (it has been done for the wetland sediments), which will make this paper much more important.

Authors' response

While thanking the Reviewer for her/his appreciation of the significance of our work, we agree that while quantification of methane-driven denitrification in freshwater systems will be the next logical step, at this point we are hesitant to scale up our results because of insufficient data.

3(b). Reviewer's Comments

Methodology

Some of the methods used by the authors are classic, e.g. isotopic N measurements. However there are many other aspects could have been better.

To date, there are only two confirmed n-damo microorganisms: *M. nitroreducens* and *M. oxyfera*. The authors tried to detect the key gene for *M. oxyfera*, but not for *M. nitroreducens*. For a paper reporting n-damo activity, this is a serious limitation.

Authors' response

We submit that the lack of information on *M. nitroreducens* is not a serious limitation. This is because while *M. nitroreducens* does oxidise methane anaerobically, it only reduces NO_3^- to NO_2^- . It does not produce N_2 on its own. Therefore it is not correct to consider *M. nitroreducens* a (nitrite-driven) N-DAMO organism like NC10 bacteria. Consequently, our observed CH_4 -stimulated N_2 production from NO_2^- cannot be due to *M. nitroreducens* as suggested. In comparison, many of the 'aerobic' Type I-III methanotrophs have various

nitrogen reducing activities and they contribute almost 14% relative abundance in our samples.

3(c). Reviewer's Comments

No methane and/to CO₂ measurement results was shown. So the paper presents evidences for nitrogen conversion, however there is no conclusive evidence for methane conversion.

Authors' response

We submit that while double labelling experiment would have demonstrated consumption of CH₄, we can think of no explanation of the effect observed by us other than CH₄ oxidation by NO₂⁻. Moreover, CO₂ may also be produced by other oxidants such as sulphate, Mn (IV) and Fe (III), present in our samples, and also by nitrate reducers that only produce NO₂⁻ (e.g. *M. nitroreducens*) but not gaseous nitrogen forms. The absence of stimulation of N₂ production by CH₄ in this coastal marine environment, where CH₄ concentrations remain very low even during anoxic periods (Naqvi et al., 2010) also shows that the effect observed by us is not an experimental artefact and is linked to the availability of CH₄.

3(d). Reviewer's Comments

Overall, the authors showed that in samples that contains no or very low level of known anaerobic methanotrophs, the addition of methane can greatly enhance denitrification. More efforts should have been made to find out the responsible microorganisms.

Authors' response

We agree. However, we submit that our observation of low numbers of NC10 bacteria is by itself quite significant because it implies that the community is more diverse than recognized so far. We show that the other denitrifying methanotrophs considered to be 'aerobic' are very abundant in anoxic waters and should play an important role in nitrogen loss.

3(e). Reviewer's Comments

No data for other important nutrients. We know AOM can couple to the reduction of sulfate, iron and manganese oxides. Also organic matters can drive denitrification. It is not essential but would be good if the authors can provide the information of these nutrients (in situ concentrations and variations during incubation tests).

Authors' response

While we agree the AOM is known to be coupled to the reduction of sulphate, iron and manganese, this study focuses on CH₄-driven denitrification, and not on AOM (please also see our response under 3b above). In fact, we have recorded high concentrations of iron and manganese in Tillari Reservoir, but we are not including these data as they do not provide any insight into the process we are investigating.

3(f). Reviewer's Comments

Other comments

I also feel the manuscript is a bit sloppy. For example,

Typos. n-dano in page 12, computed in Fig S2 caption

Authors' response

Corrected.

3(g). *Reviewer's Comments*

Sloppy sentences. e.g...More direct chemical evidence has emerged from anaerobic incubations of sediment from Lake Constance¹⁴ and of wetland soils in southeastern China¹⁰ spiked with ¹⁴CH₄ and NO₂⁻ that led to production of ¹⁴CO₂...in reference 10 paper, they used ¹³CH₄, not ¹⁴CH₄.

Authors' response

We regret the error. The statement has been corrected.

3(h). *Reviewer's Comments*

Strange numbers. 53.2±149.9 nmol, 0.071±0.076 μM; not sure you can get negative values there.

Authors' response

The high standard deviation arises from the wide range of values. It does not imply negative rates.

3(i). *Reviewer's Comments*

Wrong reference styles. e.g. ...found to be a more important pathway to N₂ (refs. 25-29)...A The pmoA and nirS genes were amplified with the primers A186-A689 (ref. 39) and nirS1F-nirS6R.

Authors' response

Corrected.

3(j). *Reviewer's Comments*

Use shorthand without defining the full term first, e.g. NIO?? OMZ.

Authors' response

Corrected.

3(k). *Reviewer's Comments*

Wrong figure caption or legends, e.g. Fig S2, legend should be NO₃+NO₂? Fig S6, caption suggest ammonium data will be presented, but not in the figure.

Authors' response

Corrected.

Reviewers' comments:

Reviewer #1 (Remarks to the Author):

I have now evaluated the revised manuscript by Dr Naqvi and colleagues. The authors addressed the comments and suggestions of the reviewers and the manuscript is much improved. Results and Discussion sections are more clear. Methods section is improved but the described addition treatments don't match those described in the Results so please revise. I trust the manuscript is acceptable for publication following some minor corrections.

Specific notes:

Line 65: "was" instead of "is". Please check if the past tense is used throughout the presentation of results.

Line 93-94: "815 measurements" of what? Please include the measured variable in the sentence.

Lines 117-118: How high was NH_4^+ in the samples where rates were measured? The presence of $^{14}\text{NH}_4^+$ will result in diluted isotopic labeling and bias the estimation of anammox.

Line 124: In the Methods section you state that DNRA was only measured in samples incubated with $^{15}\text{NO}_2^-$.

Lines 130: Again, the amendments don't match the stated in Methods section.

Lines 133-134: The isotope pairing technique is based on this assumption, if this wasn't true, ^{15}N incubation experiments would simply not be used. I agree with reviewer 3 that this amendment doesn't add much information.

Line 136: Figure 2 is not related to what is stated in the sentence.

Line 138: Did you calculate anammox in these incubations? In Figure 4 the production of $^{14}\text{N}^{15}\text{N}$ also increases which might suggest enhanced anammox since ambient $\text{NO}_3^- + \text{NO}_2^-$ at that depth is too low (Fig. 2) for $^{29}\text{N}_2$ to be produced through denitrification.

Line 150: Why didn't you estimated anammox?

Line 211: methanotrophic

Line 247: "nod gene has been detected also in the proteobacterium HdN1, along with a number of contaminated aquifers and wastewater treatment systems". Please rephrase

Line 365: you can also measure anammox from the incubations with just $^{15}\text{NO}_2^-$ when ambient $^{14}\text{NH}_4^+$ is present, as appears to be the case with your samples.

Figures: Please uniform the terms used, you have $^{15}\text{N}^{15}\text{N}$ in some figures and $^{30}\text{N}_2$ in others. You have $^{29}\text{N}_2$, $^{30}\text{N}_2$ as the axis title in Figure 4 and N_2 in Supplementary Figure 8.

Reviewer #3 (Remarks to the Author):

The quality of this manuscript has been improved significantly after revision. The initial version contained some errors, which are fixed now. Restructuring the manuscript and using table to replace text description make it much more reader-friendly, especially the methodology section. More up-to-date literature are also included now. The authors have given satisfactory answers to the majority of issues raised by the reviewers.

Reviewer #4 (Remarks to the Author):

In the manuscript "Methane stimulates massive nitrogen loss from freshwater reservoirs in India" the authors describe a large dataset on concentrations and distribution of nitrogen species in reservoirs all across India. In addition, the authors collected a large dataset on N₂ formation rates in these reservoirs. The finding of methane significantly increasing denitrification rates is novel and has to my knowledge not been reported before.

General comments

The comments of reviewer 2 have been addressed sufficiently.

Are Indian reservoirs nitrogen limited or phosphate limited? Which element determines the trophic status of freshwater in this case?

I agree with reviewer 2 that the identity of the organisms performing methane induced denitrification is missing. However, I understand that elucidating the identity of these organisms is not in the scope of this publication and would require a lot of work and time. On the other hand, I cannot follow the argumentation by the authors why classical methanotrophs or NC10 bacteria are supposed to carry out the observed denitrification. There are many questions that the authors could have addressed/discussed with the data they have available: - Are the numbers of NC10 bacteria

sufficient to explain the observed denitrification rates based on literature values of cellular activity?
– Is there evidence that the classical methanotrophs described to switch to denitrification can perform n-damo? – Was methane consumed stoichiometrically with denitrification rates? – Based on the substrate gradients measured on many occasions: is there evidence for significant n-damo in-situ?

In addition, a PCR based survey for the cited nod genes in the samples the authors collected could well be within the scope of this publication and shed light into the denitrification pathway at play in these reservoirs.

The profiles showing the distribution of nitrate/nitrite, ammonium, methane and oxygen indicate pronounced ammonia oxidation/nitrification (or leftover oxidized nitrogen species from the last mixing event, if not in steady state). Did you include nitrification measurements? How much N₂O do you expect based on ammonia oxidation?

Better comparison to “usual” concentrations (and mentioning typical ranges in similar habitats) would be helpful in all the instances where the authors describe concentrations measured in this study as “unusually low” or use other comparative expressions. How much N₂O is usually formed for a given denitrification rate?

Specific comments:

L81: when was monsoon? It's not indicated in the figures.

L89ff: What are typical values in other freshwater habitats? Are there data on other dam systems, e.g. in Europe or the US?

L112ff: please mention once which process would yield 30N₂ and which 29N₂.

L129: what do you mean with “apparent inadequacy”? Can you provide data that the measured rates are not sufficient to explain the low concentrations?

L136: What was the DOC in these samples? Is methane the only possible electron donor?

Your experiments are perfect to analyze incorporation of ¹⁵N or ¹³CH₄ and to do nano-SIMS analysis paired with FISH to identify the organisms that are active.

L187: Figure 1 shows nitrate depletion throughout the water column in spring/early summer. In this case “the very low concentrations of oxidized nitrogen forms in anoxic hypolimnia” can result from the absence of nitrate in the first place. However, I agree that for the profiles you show in Figure 2, the explanation you provide is very likely. Please specify.

L191: please provide reference values for this comparison.

L202: The pathway of nitrogen loss might be the same or similar. The organisms and the coupling to the electron donor might be different. Please be more concise.

L205ff: include methanogenesis from H₂.

L219: another explanation for the presence of aerobic methanotroph in anoxic environments is sedimentation. Aerobic methanotrophs have been found and isolated from anoxic environments repeatedly. It would be surprising if all of them thrived on n-damo.

L137: without additional evidence I would prefer a phrase like "...methylootrophs could potentially also contribute..."

L148: For a high impact publication I would have expected that the authors screen genomes and amplify the nod gene from their samples to provide additional evidence that this process plays a significant role here.

L256: You discuss that NC10 might be important (0.003-0.022% of sequences) but exclude significant anammox with (<0.013%) of the reads. Please provide a reasoning why similarly abundant microbes play a role or not or report a value better suitable for this statement.

L259ff: The community does not HAVE to rely on M. nitroreducens like archaea, but is could still be an option. All the statements following this sentence do not exclude significant participation of M. nitroreducens in the process.

L298: the stratification in 2010 shown in Fig 1 seems to be very stable. Did you see a prevalence of canonical denitrification in these samples to corroborate your statement?

Supplementary table 3: These data are interesting, but in the current form useless for the reader. Because all data of one location are mixed, it is not possible to see whether NO₂⁻ is present in anoxic or oxic parts and where N₂O was measured.

Supplementary figure 4: Same as in 3: the information is useless, because all samples might be reported here and denitrification is not expected in some of them, e.g. oxic samples. What is this table supposed to show?

Supplementary figure 4 & 5: These figures look very messy. Even with colors, it is not easy to distinguish the different profiles, especially the low concentrations.

Revision of the manuscript NCOMMS-16-27798A, "Methane stimulates massive nitrogen loss from freshwater reservoirs in India"

The manuscript has been revised taking into account all of the comments/observations made and suggestions given by Reviewer #1 and Reviewer #4. The changes made and responses to specific comments (in italics) are as follows:

REVIEWER #1

I have now evaluated the revised manuscript by Dr Naqvi and colleagues. The authors addressed the comments and suggestions of the reviewers and the manuscript is much improved. Results and Discussion sections are more clear. Methods section is improved but the described addition treatments don't match those described in the Results so please revise. I trust the manuscript is acceptable for publication following some minor corrections.

Authors' Response: We thank the Reviewer for his positive comments. The minor corrections s/he has asked for have been made as detailed below:

Specific notes:

Line 65: "was" instead of "is". Please check if the past tense is used throughout the presentation of results.

Authors' Response: Changed as suggested.

Line 93-94: "815 measurements" of what? Please include the measured variable in the sentence.

Authors' Response: "of NO₂⁻" added after "measurements" (Line 102).

Lines 117-118: How high was NH₄⁺ in the samples where rates were measured? The presence of ¹⁴NH₄⁺ will result in diluted isotopic labeling and bias the estimation of anammox.

Authors' Response: Ammonium concentration range has been included, and the text has been modified considering Reviewer's comment (Lines 129-133).

Line 124: In the Methods section you state that DNRA was only measured in samples incubated with ¹⁵NO₂⁻.

Authors' Response: This was an oversight that has been corrected (Lines 136 and 431).

Lines 130: Again, the amendments don't match the stated in Methods section.

Authors' Response: This was an oversight that has been corrected (Line 155).

Lines 133-134: The isotope pairing technique is based on this assumption, if this wasn't true, ¹⁵N incubation experiments would simply not be used. I agree with reviewer 3 that this amendment doesn't add much information.

Authors' Response: We have modified this sentence (Lines 157-159). The addition of ¹⁵NO₃⁻ + ¹⁴NO₂⁻ was done primarily to show that NO₂⁻ was being produced through NO₃⁻ reduction, and that it is quickly removed as N₂ thus explaining the near absence of NO₂⁻. We think that this does provide useful information for a system that hardly accumulates NO₂⁻.

Line 136: *Figure 2 is not related to what is stated in the sentence.*

Authors' Response: We agree. We have modified the sentence (Lines 157-159).

Line 138: *Did you calculate anammox in these incubations? In Figure 4 the production of $^{14}\text{N}^{15}\text{N}$ also increases which might suggest enhanced anammox since ambient $\text{NO}_3^- + \text{NO}_2^-$ at that depth is too low (Fig. 2) for $^{29}\text{N}_2$ to be produced through denitrification.*

Authors' Response: We have now included anammox rates from $^{15}\text{NO}_2^-$ incubations (Lines 129-132 and Table 1, and Lines 178-179 for the trips when incubations with CH_4 were made). However, in Figure 4, production of $^{29}\text{N}_2$ could not be due to anammox because incubation without CH_4 did not lead to production of $^{29}\text{N}_2$. Nitrate was present albeit in low concentration in this sample. The text has been appropriately modified (Lines 176-183).

Line 150: *Why didn't you estimated anammox?*

Authors' Response: We have now included anammox rates (Lines 178-179).

Line 211: *methanotrophic*

Authors' Response: Changed as suggested.

Line 247: *“nod gene has been detected also in the proteobacterium HdN1, along with a number of contaminated aquifers and wastewater treatment systems”.* Please rephrase

Authors' Response: Changed as suggested.

Line 365: *you can also measure anammox from the incubations with just $^{15}\text{NO}_2^-$ when ambient $^{14}\text{NH}_4^+$ is present, as appears to be the case with your samples.*

Authors' Response: As suggested, and described above, we have now included anammox rates from the $^{15}\text{NO}_2^-$ experiments.

Figures: *Please uniform the terms used, you have $^{15}\text{N}^{15}\text{N}$ in some figures and $^{30}\text{N}_2$ in others. You have $^{29}\text{N}_2$, $^{30}\text{N}_2$ as the axis title in Figure 4 and N_2 in Supplementary Figure 8.*

Authors' Response: We have now consistently used $^{14}\text{N}^{15}\text{N}$ and $^{15}\text{N}^{15}\text{N}$ in all figures.

REVIEWER #4

In the manuscript “Methane stimulates massive nitrogen loss from freshwater reservoirs in India” the authors describe a large dataset on concentrations and distribution of nitrogen species in reservoirs all across India. In addition, the authors collected a large dataset on N_2 formation rates in these reservoirs. The finding of methane significantly increasing denitrification rates is novel and has to my knowledge not been reported before.

Authors' Response: We thank the Reviewer for appreciating the significance of our work and for her/his helpful comments. In the revision we have attempted to address the issues raised by her/him, which are further detailed below against specific points.

General comments

The comments of reviewer 2 have been addressed sufficiently.

Authors' Response: We thank the Reviewer.

Are Indian reservoirs nitrogen limited or phosphate limited? Which element determines the trophic status of freshwater in this case?

Authors' Response: This question has not been explicitly addressed in the very limited published information presently available. We did not specifically look into this either.

I agree with reviewer 2 that the identity of the organisms performing methane induced denitrification is missing. However, I understand that elucidating the identity of these organisms is not in the scope of this publication and would require a lot of work and time. On the other hand, I cannot follow the argumentation by the authors why classical methanotrophs or NC10 bacteria are supposed to carry out the observed denitrification. There are many questions that the authors could have addressed/discussed with the data they have available: -Are the numbers of NC10 bacteria sufficient to explain the observed denitrification rates based on literature values of cellular activity? - Is there evidence that the classical methanotrophs described to switch to denitrification can perform n-damo? - Was methane consumed stoichiometrically with denitrification rates? - Based on the substrate gradients measured on many occasions: is there evidence for significant n-damo in-situ? In addition, a PCR based survey for the cited nod genes in the samples the authors collected could well be within the scope of this publication and shed light into the denitrification pathway at play in these reservoirs.

Authors' Response:

We attempted to identify the organisms responsible for CH₄-induced denitrification by targeting all relevant known functional genes (*pmoA* for regular aerobic methanotrophs and NC10, *mcrA* for ANME (not shown) and *nirS* for denitrifiers) at the time of sample analyses and 16S ribosomal RNA amplicon sequencing for the overall bacterial community structure. Despite such efforts, we agree with the reviewer that no definite conclusions can be drawn on the exact identity of the responsible players. However, based on the gathered data, the NC10 bacteria are so far the only known organisms that can unequivocally perform N-DAMO and they were indeed present and would certainly play a role in the N-DAMO-like process. However, their relative abundance are too low ($\leq 0.023\%$) to account for the observed denitrification rates. We have now clearly stated this in the manuscript and discuss the potential involvement of other bacteria in the methane-induced denitrification (Lines 304-310).

We did not monitor changes in CH₄ concentration, and so the question of its consumption as per the stoichiometry of N-DAMO cannot be answered. However, the distribution of NO₃⁻ and CH₄ *in situ* – i.e. low/zero NO₃⁻ in the presence of CH₄ – supports the proposed role of CH₄ in nitrogen loss (Fig. 2 a-c). We have now included another figure (Supplementary Figure 2) in the revised manuscript and added text (Lines 357-359).

Regarding the *nod* gene, the reference was cited for the sake of comprehensiveness of the discussion. However, we would like to point out that paper (ref 34) was published only a few months ago, while our sample analyses were conducted in 2011. The DNA extracts were mostly used and the remaining extracts, if any, have been frozen and thawed multiple times that we doubt it will give any reliable results if new PCRs are to be conducted on them. However, as suggested by the reviewer, we conducted searches on public databases (EMBL, NCBI) to look for related sequences similar to the *nod* genes, including both genomes and environmental metagenomes – searches for sequences related to: “Ca. Methylomirabilisoxifyfera” *nod* DAMO_2434 (CBE69496), DAMO_2437 (CBE69502), HdN1 *nod* HDN1F_02620 (CBL43845), and two *nod* sequences (KX364454 and KX364455) assembled from the metagenome of a NC10-AAA enrichment culture. However, we did not

yield any significant positive hits apart from NC10-bacterial cluster and HdN1 and those already reported by Zhu et al (ref 34), including some uncultured species from man-made bioreactor or wastewater treatment systems. Although there are some other potentially interesting hits from ‘environmental settings’ - e.g. *Kyrpidiatusciae* DSM 2912 from Solfatara in Italy (Accession number: CP002017.1), *Thermoguttaterrifontis* from a terrestrial hot spring (CP018477.1), *Thiocystisviolascens* DSM 198 from a brackish pond (CP003154.1), *Solitaleacanadensis* DSM 3403 from soil (CP003349.1), the amino acids sequence identity was too low ($\leq 36\%$) to be deduced as nitric oxide dismutase. Interestingly, one of the potential hits was a high-growth methanotrophic bacterial strain *Methylomonas* 16A(DL126916.1), though again only with low amino acids identity (34%). We have now included a sentence in lines 281-285 to indicate that we have searched for this gene in public database but yielded no significant positive hits, and neither has it been found in gamma- and alpha ‘aerobic’ methanotrophs to date.

Meanwhile, we have detected a prevalence of ‘classic aerobic’ methanotrophs, based on both functional and 16S rRNA genes. As already indicated in the last submission and current, revised manuscript (lines 255-272), nitrate reduction and N_2/N_2O production activities have indeed been demonstrated in a few methanotrophs/methylotrophs. Based on their high relative abundance, we therefore argue that they are most likely contributing to the observed denitrification. Although we cannot rule out the possibility of these organisms sinking from the overlying oxic layers to the anoxic layers, the high abundance of aerobic methanotrophs in anoxic waters is in line with growing number of reports on their presence and possible alternative lifestyles under anaerobic conditions (refs 24-28, 32). As pointed out by the reviewer, to fully pinpoint the identity of the responsible organism is beyond the scope of our study here. We have included additional reports on methanotrophs/methylotrophs (ref 32) that have the potential to respire nitrate in the revised manuscript.

We thus conclude (Lines 304-310) that the CH_4 -dependent production of N_2 from NO_2^- is best explained by the concerted activity of a diverse microbial community comprising NC10 bacteria, canonical denitrifiers, anammox bacteria and ‘aerobic’ methanotrophic alpha- and gamma-proteobacteria. Considering the low abundance of NC10 bacteria, methanotrophic alpha- and gamma-proteobacteria perhaps account for a large part of the observed effect. They could switch from respiring oxygen to nitrate, and might either produce N_2 themselves or N-compounds of intermediate redox states that are further reduced to N_2 by canonical denitrifiers or anammox bacteria.

The profiles showing the distribution of nitrate/nitrite, ammonium, methane and oxygen indicate pronounced ammonia oxidation/nitrification (or leftover oxidized nitrogen species from the last mixing event, if not in steady state). Did you include nitrification measurements? How much N_2O do you expect based on ammonia oxidation?

Authors' Response: We did not measure nitrification rate as our incubations have been sparged with helium prior to ^{15}N -label additions, but based on the available information from elsewhere we suggest that nitrification could account for the majority of the observed distribution of N_2O , with likely significant contribution from denitrification only in a few cases (please see added text in lines 364-382).

Better comparison to “usual” concentrations (and mentioning typical ranges in similar habitats) would be helpful in all the instances where the authors describe concentrations measured in this study as “unusually low” or use other comparative expressions. How much N_2O is usually formed for a given denitrification rate?

Authors' Response: We have included few additional references on N₂O in anoxic lakes/reservoirs (refs 21, 50-52), and also providing estimates of N₂O yield during denitrification, which is highly variable (0.1-6%; refs 53,54). We have discussed these in the revision (Lines 373-385) to cover the points addressed by the Reviewer.

Specific comments:

L81: when was monsoon? It's not indicated in the figures.

Authors' Response: The monsoon period is now demarcated in Figure 1.

L89ff: What are typical values in other freshwater habitats? Are there data on other dam systems, e.g. in Europe or the US?

Authors' Response: A short discussion with added information from some other systems has now been included in lines 373-381.

L112ff: please mention once which process would yield ³⁰N₂ and which ²⁹N₂.

Authors' Response: As suggested we have included a sentence (Lines 122-124) at the beginning of the short section on the sources of ²⁹N₂ and ³⁰N₂ in our incubations.

L129: what do you mean with "apparent inadequacy"? Can you provide data that the measured rates are not sufficient to explain the low concentrations?

Authors' Response: We have clarified this in the revised manuscript (see added text in lines 144-154).

L136: What was the DOC in these samples? Is methane the only possible electron donor?

Authors' Response: We did not measure DOC in these samples and do not exclude that organic carbon could be the electron donor in the experiments without added methane. However, the huge stimulation of the denitrification rates when adding methane (back) into the incubation after degassing with helium gives us confidence that most of the denitrification is associated with methane oxidation.

Your experiments are perfect to analyze incorporation of ¹⁵N or ¹³CH₄ and to do nano-SIMS analysis paired with FISH to identify the organisms that are active.

Authors' Response: These experiments were done in 2011-2012. We did not take sample for CARD-FISH/nanoSIMS, nor did we do our incubations with ¹³CH₄.

L187: Figure 1 shows nitrate depletion throughout the water column in spring/early summer. In this case "the very low concentrations of oxidized nitrogen forms in anoxic hypolimnia" can result from the absence of nitrate in the first place. However, I agree that for the profiles you show in Figure 2, the explanation you provide is very likely. Please specify.

Authors' Response: We now discuss that low NO₃⁻ concentrations in anoxic waters are not due to an absence of nitrate in the first place (Lines 83-91).

L191: please provide reference values for this comparison.

Authors' Response: A comparison with nitrogen loss rate from a recent study (ref 21) has been made in lines 151-154. This has already been addressed above (Lines 364-383).

L202: The pathway of nitrogen loss might be the same or similar. The organisms and the coupling to the electron donor might be different. Please be more concise.

Authors' Response: This sentence has now been deleted for reasons mentioned below.

L205ff: include methanogenesis from H₂.

Authors' Response: We agree with the reviewer that H₂ could be added here. However, after reconsidering this section we found that a correct and justified discussion on the sources of methane and its regulation in lakes vs. marine environments would be lengthy and complex. We instead deleted this statement as it is beyond the scope of our manuscript dealing with methane oxidation and denitrification.

L219: another explanation for the presence of aerobic methanotroph in anoxic environments is sedimentation. Aerobic methanotrophs have been found and isolated from anoxic environments repeatedly. It would be surprising if all of them thrived on n-damo.

Authors' Response: We would like to clarify that we did not imply that ALL of them were denitrifying methanotrophs, but instead, we argue that at least a number of them would contribute to the observed effect, either as individual organisms (N-DAMO) or in combined effort with others. We have now clarified this in the discussion in text (Lines 304-310). Please also refer to response to earlier comments above.

L137: without additional evidence I would prefer a phrase like "...methylophs could potentially also contribute..."

Authors' Response: We have added discussion elsewhere to propose a more important role of conventional methanotrophs in denitrification. The modified sentence (Line 271) is consistent with this view.

L148: For a high impact publication I would have expected that the authors screen genomes and amplify the nod gene from their samples to provide additional evidence that this process plays a significant role here.

Authors' Response: We have screened genomes/metagenomes in public data bases as suggested that yielded no positive hits, but did not amplify the nod gene (please see response to earlier comments above). More importantly, we do not argue that organisms capable of dismutating nitric oxide to O₂ and N₂ (e.g. members of the NC10 phylum) are the major players responsible for the observed methane-induced denitrification, but a rather concerted effort from a number of potential players (see response to earlier comment).

L256: You discuss that NC10 might be important (0.003-0.022% of sequences) but exclude significant anammox with (<0.013%) of the reads. Please provide a reasoning why similarly abundant microbes play a role or not or report a value better suitable for this statement.

Authors' Response: Based on the comments by the reviewer (see above), we have now revised the discussion regarding the importance of NC10. We also clarify that we do not rule out the involvement of anammox, despite the generally low rates (Lines 293-296).

L259ff: The community does not HAVE to rely on M. nitroreducens like archaea, but is could still be an option. All the statements following this sentence do not exclude significant participation of M. nitroreducens in the process.

Authors' Response: We agree that *M. nitroreducens* might be contributing to nitrate reduction to nitrite along with other potential nitrate reducers present, including a number of aerobic methanotrophs which were more abundant than *M. nitroreducens*. We have modified the sentence slightly (Line 298) to accommodate the Reviewer's view.

L298: the stratification in 2010 shown in Fig 1 seems to be very stable. Did you see a prevalence of canonical denitrification in these samples to corroborate your statement?

Authors' Response: We did not see much denitrification in Tillari although on a few occasions we observed accumulation of N_2O in high concentrations that we believe would be produced through denitrification. On these occasions denitrification rates were not measured.

Supplementary table 3: These data are interesting, but in the current form useless for the reader. Because all data of one location are mixed, it is not possible to see whether NO_2^- is present in anoxic or oxic parts and where N_2O was measured.

Authors' Response: We now provide data separately for the epilimnion and hypolimnion of the reservoirs in the revised Table 3.

Supplementary table 4: Same as in 3: the information is useless, because all samples might be reported here and denitrification is not expected in some of them, e.g. oxic samples. What is this table supposed to show?

Authors' Response: We only performed rate measurements in anoxic samples (Lines 115-117), except in one case (Markandeya in January 2012; lines 156-157). This table was added on the recommendation of one of the referees.

Supplementary figure 4 & 5: These figures look very messy. Even with colors, it is not easy to distinguish the different profiles, especially the low concentrations.

Authors' Response: We admit that it takes some effort to digest the data presented in these figures. The problem is that there is too much data, but we have tried to do our best to present them as clearly as we could.

REVIEWERS' COMMENTS:

Reviewer #1 (Remarks to the Author):

The authors addressed all the comments and suggestions of the reviewers. I trust that the manuscript is now acceptable for publication.

Reviewer #4 (Remarks to the Author):

The authors have sufficiently addressed all my comments. Thank you very much!

Response to Reviews

Since the reviewers have not made any comments on the last version, no response is needed. We wish to thank them for their time and extremely helpful comments.